# A Risk Assessment Framework of Hybrid Offshore Wind–Solar PV Power Plants under a Probabilistic Linguistic Environment

**Qinghua Mao** , **Mengxin Guo, Jian Lv ** , **Jinjin Chen, Pengzhen Xie and Meng Li**

School of Economics and Management, Yanshan University, Qinhuangdao 066004, China;
maoqh@ysu.edu.cn (Q.M.); guomx1207@163.com (M.G.); cjj_12342021@163.com (J.C.);
xiepengzhen123@163.com (P.X.); lemmon0922@163.com (M.L.)
\* Correspondence: lvjian2023@163.com; Tel.: +86-155-2801-5320

**Abstract:** Hybrid offshore wind–solar PV power plants have attracted much attention in recent years due to its advantages of saving land resources, high energy efficiency, high power generation efficiency, and stable power output. However, due to the project still being in its infancy, investors will face a series of risks. Hence, a multi-criteria group decision-making framework for hybrid offshore wind–solar PV power plants risk assessment is constructed in this paper. Firstly, 19 risk indicators are identified and divided into five groups. Secondly, probabilistic linguistic term sets are then introduced to evaluate the criteria values to depict uncertainty and fuzziness. Thirdly, the expert weight determination model is built by combining subjective and objective weights based on expert information, the entropy and interaction-entropy measures of probabilistic linguistic term sets. Fourthly, the expert evaluation information is aggregated by transforming probabilistic linguistic term sets into triangular fuzzy numbers based on generalized weighted ordered weighted averaging operator. Additionally, the risk level is determined using the fuzzy synthetic evaluation method. Finally, the proposed method is applied to a case study and the risk level is slightly high with the similarity measure result of 0.938. Then, the risk indicator system and corresponding countermeasures can provide scientific reference for investment decisions and risk prevention.

**Keywords:** hybrid offshore wind–solar PV power generation; risk assessment; probabilistic linguistic term sets; triangular fuzzy numbers; fuzzy synthetic evaluation method

## 1. Introduction

Due to the serious consequences of the greenhouse effect, global warming is increasing and global energy reform is imperative to improve environmental issues. In addition, with coal, oil, natural gas, and other non-renewable energy storage greatly reduced, the effective use of renewable energy is extremely urgent. Renewable energy, especially wind and solar, are two potential candidates to remove carbon footprints, which are cleaner and safer. In recent years, the penetration of wind and solar resources has increased. The scarcity of habitable land encourages the development of renewable energy projects in the marine environment. With the progress of technology and the reasonable utilization of existing resources, the construction of offshore wind power stations and offshore solar power stations shows a significant growth trend. As for wind energy, as a kind of renewable and clean energy, offshore wind power occupies a certain proportion of the energy industry and plays a crucial role in relieving energy pressure [1]. As for solar energy, floating photovoltaic (FPV) systems are the core of offshore photovoltaic power generation. The main advantage of FPV systems is the water cooling on the solar cells [2]. This effect results in a higher energy conversion efficiency of the floating panels, which can generate up to 10% more electricity [3]. In addition, it has the advantages of inhibition of algae reproduction, convenient cleaning of PV equipment, and so on [4].

Meanwhile, as an energy complementary system to provide more stable and flexible power output, hybrid offshore wind–solar PV power generation has attracted much

attention in recent years, which has realized the expansion of energy utilization in the space–time dimension. A basic arrangement of hybrid offshore wind–solar PV power generation would be filling with FPV panels the free-surface amidst the offshore wind turbines, which ensures smooth output of power generation on the basis of reasonable utilization of available sea area. Additionally, this arrangement also avoids interferences in the production of both renewables [5]. Mario López et al. [5] assessed the potential for developing floating wind and solar energy off the coast of Asturias and demonstrated a production synergy of offshore wind–solar farms. Compared with a typical offshore wind farm, the capacity of the hybrid offshore wind–solar PV power farm is 10 times higher, and the amount of electricity generated per surface area is seven times higher. In this way, the utilization of marine space is optimized, while at the same time, the power output is significantly smooth. It should be considered in future marine renewable energy projects.

However, the hybrid offshore wind–solar PV power generation project has just been studied by scholars, and the relevant core technologies are not strong enough, including seawater corrosion treatment, optimal array layout, power grid connection technology, converters, etc. In addition, the market environment is not stable enough. There is very little information to refer to for the project. In terms of these challenges, investors and project managers will inevitably encounter some risks during the period of construction and operation of the hybrid offshore wind–solar PV power generation project. Moreover, energy projects have the characteristics of large capital input, strong unity, and long cycle; risk management is particularly necessary in this case. Hence, reasonable risk assessment and scientific risks countermeasures play a significant role in the whole life cycle of the project. However, this problem has not been widely studied by scholars. It is obvious that there is a lack of comprehensive risk assessment framework and a related risk indicator system for the hybrid offshore wind–solar PV power project. Therefore, the meaning of the research in this paper cannot be ignored.

Up to now, there is little literature on risk assessment for the hybrid offshore wind–solar PV power projects, but the risk assessment has been implemented for other offshore renewable energy projects. For offshore wind projects, Dai, Ehlers et al. [6] analyzed the risk of collision between service vessels and offshore wind turbines and concluded that collisions between turbines and service vessels even at low speed may cause structural damage to the turbines; accordingly, certain risk responses were put forward in the aspect of design. Gatzert et al. [7] analyzed the risk and corresponding risk management both for onshore and offshore wind projects. Hong and Moller [8] focused on the economic risk assessment of offshore wind farms from the perspective of tropical cyclones. Snyder and Kaiser [9] studied the offshore wind plant using ecological and economic analysis. Shafiee and Dinmohammadi [10] comparatively analyzed the risks of onshore wind power and offshore wind power and proposed that both systems faced many of the same risks; however, there are some main differences worth considering. For photovoltaic power generation projects, Trapani et al. [11] proposed an alternative to flexible thin-film photovoltaics that floats on the water and focuses on technical and economical assessment of offshore PV systems at the waterline. Wu et al. [12] proposed a three-phase risk assessment model for the PV poverty alleviation projects. Kayser et al. [13] identified the most critical risk factors currently hindering the development of China's PV projects using the improved Delphi method. Wu et al. [14] established a risk assessment framework of offshore PV projects based on the MAGDM method. A criteria system was built including risk factors in macro-economic, technical, environmental, and management aspects. A fuzzy comprehensive evaluation model was also constructed by combining the hesitant linguistic fuzzy sets, triangular fuzzy sets, and analytic network process (ANP) methods. Gao et al. [15] considered market risks, established an index system including economic, technical, environmental, and market risk factors, and established a comprehensive risk assessment framework for offshore PV projects under probabilistic language term sets. Considering the uncertainty of processing information and the confidence of expert judgement, Zhou et al. [16] established a risk assessment model of China's offshore photovoltaic power generation projects with D-

number and ANP method. In addition, Wu et al. [17] established a risk assessment model of an offshore wave–wind–solar–compressed air energy storage power plant based on the fuzzy comprehensive evaluation method. For the hybrid offshore wind–solar PV power projects, Mario López et al. [5] assessed the potential for developing floating wind and solar energy off the coast of Asturias and demonstrated a production synergy of offshore wind–solar farms. Syed Raahat Ara et al. [18] proposed a two-level planning approach to analyze techno-economic feasibility of hybrid offshore wind–solar PV power plants. Dhunny, A Z studied the site selection of hybrid onshore wind–solar PV power plants [19]. Figure 1 illustrates the analysis of the research status of the hybrid offshore wind–solar PV power plant and the comparison with other offshore new energy power plants. It is obvious that risk assessment of hybrid offshore wind–solar PV power generation projects has not been extensively studied. Hence, this paper will draw on the research published on other energy projects to establish a targeted and practical indicator system and a comprehensive risk assessment framework suitable for hybrid offshore wind–solar PV power generation projects.

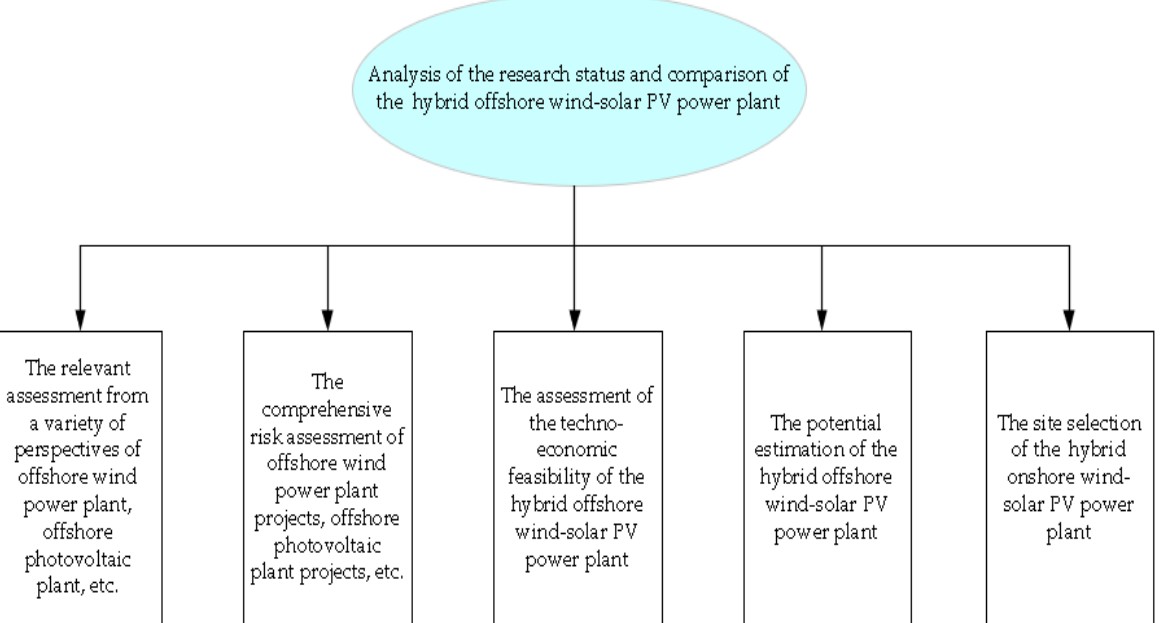

**Figure 1.** The analysis of the research status and comparison of the offshore new energy power.

This paper aims to establish a practical indicator system for the risk assessment of the hybrid offshore wind–solar PV power generation project and develop an effective comprehensive risk evaluation framework to evaluate the risk level of the hybrid offshore wind–solar PV power generation project for related management personnel. The contributions of the proposed method can be briefly summarized as follows: (1) Through literature review, case study, referring to the risk assessment of previous energy projects, and inviting relevant experts, the expert committee was established, and finally the expert committee established the risk indicator system. (2) The probabilistic linguistic term sets (PLTSs) are introduced in this paper to describe the risk assessment information. Compared with other forms of fuzzy sets, PLTSs not only allow experts to express preferences in different linguistic terms but also give corresponding probability information for each linguistic term, which can better retain the original evaluation information. (3) In this paper, a subjective and objective weighting method is used to determine expert weight. The subjective weight is determined according to the expert's position, working time, project experience and other standards, and objective weight based on the entropy and interaction-entropy measures of PLTSs. The combination of subjective and objective weights is more in line with the actual decision-making environment. (4) The expert evaluation information is aggregated

by transforming PLTSs into triangular fuzzy numbers based on the generalized weighted ordered weighted averaging (GWOWA) operator [20], which uses the original evaluation information effectively to the maximum extent. Additionally, the risk level of the project is determined by fuzzy synthetic evaluation method and similarity measures and provides the corresponding risk response strategy.

The rest of the study is as follows. Section 2 reviews the PLTSs, TFN, and FSE methods, which will be used in this paper. Section 3 analyzes the risk factors, and a comprehensive indicator system for the hybrid offshore wind–solar PV power generation projects is developed. A decision framework of the risk assessment for the hybrid offshore wind–solar PV power generation projects is presented in Section 4. A case study, sensitivity analysis, and comparative analysis are presented in Section 5. We present a discussion and offer corresponding risks countermeasures in Section 6. Finally, Section 7 concludes this paper.

## 2. Literature Review

The multiple-attribute group decision-making (MAGDM) problem is a sub-discipline of operations research. The MAGDM theory has received extensive attention and quite a few achievements have been presented in the past decades. Because of the variety of risk factors, it is clear that the risk assessment of hybrid offshore wind–solar PV power generation projects is a typical MAGDM problem. Two main reasons for the uncertainty of decision information were determined. Firstly, risk assessment of a project is universally conducted in the planning and feasibility study stage. However, the risk assessment is only an estimate of what will happen in the future. Therefore, there is uncertainty in the process of risk assessment. Secondly, the judgements and evaluations are entirely dependent on the experience and knowledge of DMs in the risk assessment. However, it is clear that DMs are not entirely rational, so fuzziness exists. Therefore, the MAGDM method is often an effective tool for these problems with imperfect, vague, and imprecise information, which plays a significant role in the reasonableness and accuracy of risk assessment. In 1965, Zadeh [21] proposed the concept of fuzzy sets to express uncertainty and fuzzy information. Compared with specific numbers, fuzzy sets can express the uncertainty of objective things and the fuzziness of subjective cognition better and have been widely used to solve these problems. On this basis, some scholars have developed the fuzzy sets from different angles and put forward its extended forms including type-2 fuzzy sets [22], intuitionistic fuzzy sets [23], interval fuzzy sets [23], hesitant fuzzy sets [24], hesitant fuzzy language terms [25], and PLTSs etc. [26]. The probabilistic linguistic term sets (PLTSs) are introduced in this paper to describe the risk assessment information. PLTSs allow experts to express preferences with multiple different linguistic terms and give corresponding probability information for each term. For example, when assessing the market risk of hybrid offshore wind–solar PV power generation projects, an expert may express that he/she is 60% sure that it is slightly high ($s_4$), and 40% sure that it is medium ($s_3$); this evaluation information can be expressed in the form of PLTS $\{s_4(0.6), s_3(0.4)\}$. It is obvious that PLTS can better retain the original evaluation information. In addition, in terms of expert evaluation information aggregation, PLTSs are transformed into triangular fuzzy numbers based on a generalized weighted ordered weighted average (GWOWA) operator. The triangular fuzzy number proposed by Zadeh [21] has been widely used in MAGDM problems for the purpose of making decisions more in line with a realistic decision environment [10]. An inexact interval-valued triangular fuzzy based multi-attribute preference framework was conducted by Ren et al. [27]. It mainly takes vagueness in parameter values into account. Some hesitant triangular fuzzy aggregation operators were proposed by Zhao et al. [28]. Additionally, they investigated their application to MAGDM problems, and an illustrative example was used to show the validity of these operators. It can be seen clearly from the above literature that the TFN is an effective tool for dealing with the MAGDM problem with uncertain information and relatively simple calculation process well. Therefore, the triangle fuzzy number was introduced by many researchers to solve risk assessment and

performance analysis. In addition, some scholars have studied the methods of formal transformation [20,29], which play an important role in MAGDM.

At present, for the MAGDM problem, many decision methods have been studied extensively. TOPSIS (Technique for Order Preference by Similarity to an Ideal Solution), VIKOR (VIseKriterijumska Optimizacija I Kompromisno Resenje), ELECTRE (Elimination and Choice Translating Reality), and TODIM (an acronym in Portuguese for interactive and multicriteria decision making) methods have been widely studied and applied by scholars. Chen, et al. [30] conducted a hybrid MCDM approach based on ANP-entropy TOPSIS to solve the material supplier selection problem. Li, et al. [31] proposed an MCDM model based on the grey correlation and TOPSIS under interval-valued intuitionistic fuzzy environment to select the cooperative partner in military–civilian scientific and technological collaborative innovation. Tufail, et al. [32] studied the VIKOR method for MCDM based on a bipolar fuzzy soft $\beta$-covering-based bipolar fuzzy rough set model. Additionally, the proposed method can solve the site selection for solar power plants well. Peng, et al. [33] conducted an integrated decision support model based on regret theory and ELECTRE III to solve the problem of investment risk evaluation for new energy resources. Wu, et al. [34] conducted a framework for offshore wind power station site selection based on ELECTRE III under an intuitionistic fuzzy environment. Zhang, et al. [35] proposed the Wasserstein distance-based probabilistic linguistic TODIM method, which can solve the problem of the evaluation of sustainable rural tourism potential. Ding, et al. [36] studied the interval-valued hesitant fuzzy TODIM method for dynamic emergency responses. However, the above method can be utilized since these methods are mainly applied in optimal selection or decision making for multiple potential projects instead of one targeted project. Fortunately, the fuzzy synthetic evaluation method (FSE) can solve this problem as an effective and practical approach for evaluating targeted non-deterministic problems with qualitative languages through membership grade theory [37]. FSE provides a method of expression and definition of fuzzy variables in mathematical logic, which can be utilized in quantifying risk level, severity, and the impact of fuzzy risk variables. Therefore, this method can be used to not only express the empirical knowledge of project managers but also help draw reliable decisions from fuzzy facts via language definitions [38]. Many researchers have adopted this method for risk assessment, investment decision making, site selection, and so on. Wu, Li [37] evaluated the risk level of China's PPP straw power generation project via the FSE model. Some scholars also evaluated the real estate investment risk through the FSE model. Additionally, they verified the scientific nature and practicality of the model adopted. Wu, Jia [39] analyzed the advantages of the FSE model and assessed the risk of the electric vehicle supply chain based on this method. Moreover, Wu, Li [14] selected the FSE model to assess China's offshore photovoltaic power generation projects based on the FSE model.

Thus, this paper will build a comprehensive risk assessment framework of a hybrid offshore wind–solar PV power generation project with the advantages of the entropy and interaction-entropy measures of PLTSs, GWOWA operator, and the FSE method based on the above related scientific research.

## 3. The Construction of the Indicator System of Risk Assessment on a Hybrid Offshore Wind–Solar PV Power Generation Project

It is an essential prerequisite for investors to identify risk factors to conduct risk management [40]. In this study, the literature review method was first adopted and databases such as the web of science and Elsevier were used. Moreover, literature published by domestic and foreign scholars were searched using keywords such as offshore wind power, offshore photovoltaic, and offshore new energy projects. We obtained extensive articles on the risk assessment of these projects [6–10,14–17]. At the same time, the risk factors can be collected by referring to the conference reports of offshore new energy projects and case studies of existing offshore new energy projects. Through the above activities, we obtained a large number of risk indicators of offshore new energy power

generation projects. Then, three experts with rich experience in offshore energy projects were invited to set up an expert committee. The expert committee selected and classified the large collection of risk indicators according to their frequency and weight in published articles. Finally, the expert committee determined the risk criteria system of the hybrid offshore wind–solar PV power generation project through repeated discussion, 19 risk factors were identified, and they were divided into five groups. The risk criteria system is shown in Figure 2.

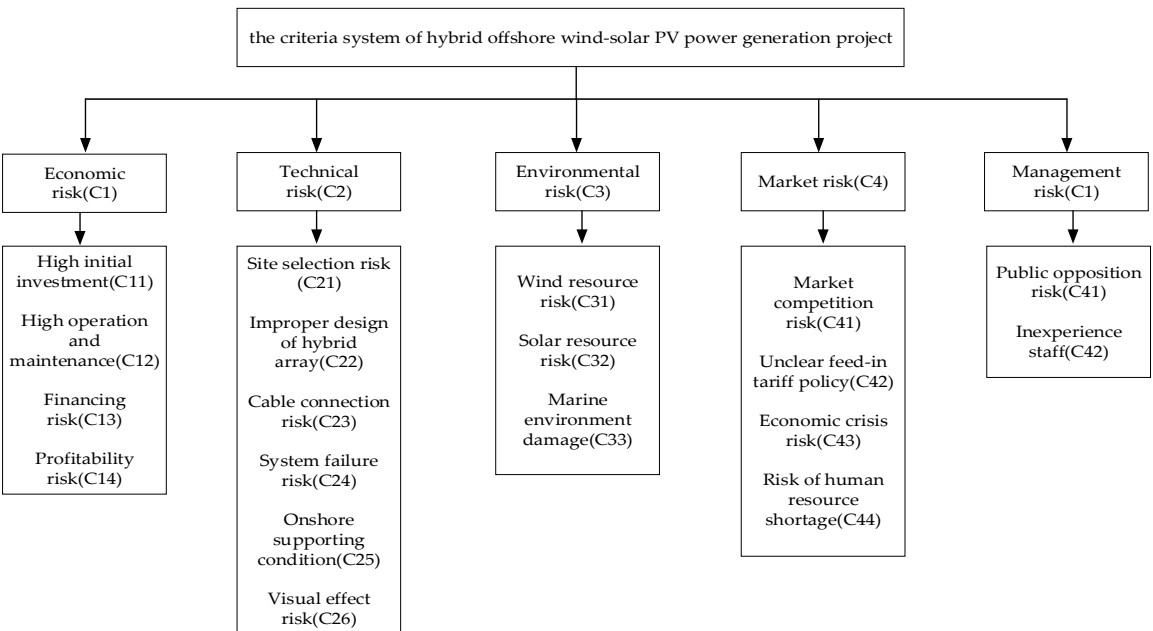

**Figure 2.** The criteria system of the hybrid offshore wind–solar PV power generation project.

### 3.1. Economic Risk (C1)

High initial investment (C11): The initial investment of the project mainly includes equipment purchase cost, installation cost, construction cost, reserve cost, interest during the construction period, etc. Hybrid offshore wind–solar PV power generation projects will face higher costs than other energy projects, given the complexity of design and manufacturing processes due to higher performance requirements. For example, underwater cable requires high construction technology and expensive equipment, which increases the financial pressure of the investment of the project.

High operation and maintenance costs (C12): The construction and operation of renewable energy plants is critical to project management. Wind turbines and photovoltaic equipment are prone to deformation, metal corrosion, material aging, and other risks due to sea salt corrosion and sea breeze intrusion. Operation and maintenance costs increase gradually as the cycle continues.

Financing risk (C13): A relatively large capital scale is obviously required for hybrid offshore wind–solar PV power generation projects. Hence, the financing process plays an important role in the smooth implementation of the project. Financing risk mainly consists of the uncertainty generated by financing activities, such as financing guarantee, financing structure design, and financing channel selection [41]. The technology of hybrid offshore wind–solar PV power generation projects is still not mature; there may be significant obstacles and risks in the financing process.

Profitability risk (C14): The profitability of a project refers to the ability to increase the value of the funds invested in the project, which is easily affected by interest rate changes, inflation, capital turnover, and other problems, leading to the reduction in the investment benefit of the enterprise. The profitability risk of hybrid offshore wind–solar PV power generation projects should be paid more attention under the background of government subsidy withdrawal. In order to conform to reality, considering the time value of capital

and reflecting the economic effect of the project, the net present value is usually the main factor, supplemented by internal rate of return and dynamic payback period, and the profitability risk of the project is evaluated.

### 3.2. Technical Risk (C2)

Site selection risk (C21): Site selection is the key part in hybrid offshore wind–solar PV power generation projects, involved in solar resources, wind resources, distance to the center of the load, geological conditions, etc. In addition, improper site selection not only has a negative impact on project earnings, but also can cause problems such as not starting construction as scheduled; therefore, has a certain risk.

Improper design of hybrid array (C22): The array design of hybrid offshore wind–so lar PV power generation mainly includes the arrangement of wind turbines and photovoltaic panels, which needs to consider the wake effect of wind turbines and the shadow effect of solar photovoltaic panels. Unreasonable array design may lead to inadequate resource utilization and low power generation efficiency.

Cable connection risk (C23): Under the gravitational action of celestial bodies, the sea water in coastal areas has periodic fluctuations, known as oceanic tides. This poses a risk for this project, where the downward pull of seawater pulls the cables as they fall back. If the inverter cable connection and node connection between the PV panel and turbine are not designed properly, the cable will not be able to cope with the effect of tide.

System failure risk (C24): The arrangement of the hybrid offshore wind–solar PV power station may have a certain impact on birds, ships, and other original routes, and the collision between turbines and birds or ships will inevitably lead to the failure of the power generation system, which has certain risks.

Onshore supporting condition risk (C25): Onshore support conditions refer to the important factors for project construction, operation, such as transportation conditions and power transmission, and distribution systems. Hence, traffic conditions should be taken into account because of the impact on the transport of large equipment. In addition, we also need to consider whether the onshore grid and future plans can meet the support needs.

Visual effect risk (C26): Wind turbines are likely to pose a threat to bird species by visually affecting flight or migration. In addition, although the tempered glass of photovoltaic modules has a high light transmittance, it still cannot completely avoid the reflection phenomenon, which may cause a visual impact on coastal residents. Large-scale hybrid offshore wind–solar PV power stations may also have a visual impact on the coastal landscape.

### 3.3. Environmental Risk (C3)

Wind resource risk (C31): The periodic variation in wind speed and wind resources is vital for offshore wind power generation. To evaluate the development potential of offshore wind power generation in an area in the early stage of a project has a profound impact on the long-term development of the project.

Solar resource risk (C32): Whether there is ample photovoltaic power generation of solar energy resources or not is quite important. Atmospheric haze, dust, and other obstacles may reduce the power output of photovoltaic power generation. In addition, the change in climate may also be caused by the long-term prediction of solar energy resources; electric power systems may not be able to achieve the desired output, which will affect the project's profit.

Marine ecological damage (C33): The hybrid wind–solar PV power generation plants have large scales, which will inevitably affect the marine ecological environment in the development and construction processes. For example, the laying of submarine transmission cables will make seabed sediments float and thus influence the reproduction of plankton. In addition, the coastal habitat of birds will be inevitably occupied, and their nests will be affected. Due to this damage to the marine ecosystem, environmental protection agencies or environmentalists may raise objections.

*3.4. Market Risk (C4)*

Market competition risk (C41): The competitiveness of the market will directly determine the viability of it in renewable power projects. The competition of hybrid offshore wind–solar PV power plants includes offshore wind, offshore photovoltaic, and other offshore renewable energy hybrid projects. Because the hybrid wind–solar PV power generation project is still in the begin stage, its research and development ability are relatively weak, and it lacks marketing ability and has certain technical limitations; therefore, there is a risk of being replaced in the market.

Unclear feed-in tariff policy (C42): Pricing policy is critical to project profitability, but there is no clear pricing policy for hybrid wind–solar PV power generation projects in the energy project market, which will bring uncertainty to project revenue.

Economic crisis risk (C43): Natural disasters and other force majeure events may have a certain influence on the market economy. For example, many enterprises go bankrupt and unemployment rate increases to induce economic crises, which may bring risks such as capital chain fracture to the project and make the normal operation of the project impossible.

Risk of human resource shortage (C44): At present, in the context of COVID-19, a public health emergency, the economic benefits and human resource management of enterprises have received a great impact. Many enterprises are faced with situations of layoffs and recruitment difficulties. For the hybrid wind–solar PV power generation project, relevant personnel should have certain technical requirements, which makes it harder to recruit staff.

*3.5. Management Risk (C5)*

Public opposition risk (C51): The public's attitude towards the power station has a great influence on the successful implementation of the power station. However, the noise and occupancy of the hybrid power station may cause certain restrictions on the life of the surrounding residents.

Inexperienced staff (C52): Hybrid wind–solar PV power generation projects are still in the begin stage, and building, installing, and operating them in a marine environment requires extensive expertise and work experience, which is somewhat risky.

## 4. The Risk Assessment Framework of Hybrid Wind–Solar PV Power Generation Projects

**Step 1. Determination of the weight of risk indicators**

In the risk assessment of the project, the importance of each criterion is different, so the importance of each criterion needs weight to reflect it. At the same time, there is a certain correlation among these criteria. For example, improper design of a hybrid array (C22) and unreasonable site selection (C21) of the hybrid power plant will increase project operation and maintenance costs (C12). Considering this situation, the ANP method is used in this paper to determine the weight of attributes. Firstly, the expert group analyzed and determined the intrinsic dependence among the criteria. Secondly, we used a 1–9 scale method to determine the risk degree of each criterion and made pairwise comparisons to obtain the judgment matrix. After calculating the judgment matrix, the unweighted super matrix can be obtained. Thirdly, the influence matrix of the index group was obtained by comparing the relationships between the index groups. Finally, the super weighted matrix and limit matrix were obtained using software calculation, and the global weight and local weight of the indicator were obtained [14].

**Step 2. Defining the PLTS and obtaining the evaluation from experts**

Considering that there are many factors involved in risk assessment of hybrid wind–solar PV power generation projects, experts may use more than one language term for evaluation and may hesitate among several language terms. In addition, experts may have different preferences for each language term in expert evaluation. Therefore, in this paper, probabilistic language term sets (PLTSs) were used to give criteria evaluation information to reduce information loss and improve the accuracy of evaluation results.

**Theorem 1.** *Ref. [26] Let* $S = \{s_i | i = 0, 1, \cdots, 2\tau\}$ *be a linguistic term set, a PLTS is defined as:*

$$L(p) = \left\{ L^k(p^k) | L^k \in S, p^k \geq 0, k = 1, 2, \cdots, \#L(p), \sum_{k=1}^{\#L(p)} p^k \leq 1 \right\} \tag{1}$$

where $L^k(p^k)$ is the linguistic term $L^k$ associated with probability $p^k$, and $\#L(p)$ is the number of all linguistic terms in $L(p)$.

**Example 1.** $S = \{s_t | t = -3, -2, -1, 0, 1, 2, 3\}$ *be an LTS.* $L_1(p) = \{(s_{-3}, 0.2), (s_{-2}, 0.8)\}$ *and* $L_2(p) = \{(s_{-2}, 0.4), (s_0, 0.3), (s_1, 0.2)\}$ *are two PLTSs.*

We can see from example 1 that, in some cases, the sum of the probabilities of $L_2(p)$ is less than 1. The elements in the two PLTSs are not equal. Hence, the method of PLTS normalization is defined as follows.

**Theorem 2.** *Ref. [26] Let* $L(p) = \left\{ L^k(p^k) | L^k \in S, p^k \geq 0, k = 1, 2, \cdots, \#L(p), \sum_{k=1}^{\#L(p)} p^k \prec 1 \right\}$ *be a PLTS. Then the normalized PLTS for* $L(p)$ *is defined as:*

$$L(p) = \{L^{(k)}(p\prime(k)) | k = 1, 2, \cdots, \#L(p)\} \tag{2}$$

where $p\prime(k) = p(k) / \sum_{k=1}^{\#L(p)} p(k)$ for $k = 1, 2, \cdots, \#L(p)$.

If the number of elements of two PLTSs is not equal, we need to add the $|\#L_1(p) - \#L_2(p)|$ element to the few PLTS; the element added is the smallest in PLTS and the probability value is 0. $L_1(p) = \{(s_{-3}, 0.2), (s_{-2}, 0.8), (s_{-3}, 0)\}$ and $L_2(p) = \{(s_{-2}, 0.44), (s_0, 0.33), (s_1, 0.22)\}$ are obtained.

**Step 3. Determining the weight of experts**

The weight of experts is determined by combining subjective and objective weights, and the subjective weighting method uses four criteria to distinguish the weight of experts: position held, project experience, education level, and working time, which is shown in Table 1. The score of each expert is obtained according to the four standards, and the subjective weight of the expert is obtained according to the score. The objective weighting method is based on the combination of probabilistic language entropy and interaction-entropy measures. The final weight is the average of subjective and objective weights.

**Table 1.** The score of expert subjective weight based on the information of experts.

| Criterion | Classification | Score |
|---|---|---|
| Position | senior scholars | 4 |
| | primary scholars | 3 |
| | engineer | 2 |
| | technician | 1 |
| Project experience | more than 7 | 3 |
| | 3–7 | 2 |
| | less than 3 | 1 |
| Education level | doctor and above | 3 |
| | master | 2 |
| | undergraduate and below | 1 |
| Working time | more than 7 years | 3 |
| | 3–7 years | 2 |
| | less than 3 years | 1 |

The subjective weight of expert $E_i$:

$$\omega(E_i) = \frac{s(E_i)}{\sum_{i=1}^{k} s(E_i)} \tag{3}$$

1.　Probabilistic linguistic entropy measure

If the uncertainty degree of decision information given by expert $E_i$ is greater, the corresponding entropy measure will be larger, indicating that the useful information provided by expert $E_i$ is less. In this case, expert $E_i$ should be given a smaller weight [15].

$$\overline{\omega_i} = \frac{\sum_{j=1}^{n} (1 - E(L_{ij}(p)))}{\sum_{i=1}^{m} \sum_{j=1}^{n} (1 - E(L_{ij}(p)))} (i = 1, 2, \cdots, m) \tag{4}$$

where $E(L_{ij}(p))$ is the entropy of PLTS $L_{ij}(p)$. It is defined as follows.

**Theorem 3.** *Ref.* [15] *Let* $L(p) = \{L^k(p^k) | k = 1, 2, \cdots, \#L(P)\}$ *is a PLTS. The entropy measure of PLTS is:*

$$E(L(p)) \begin{cases} \sum_{i=1}^{\#L(p)} \sum_{j=1}^{\#L(p)} 4 p_i p_j \cdot f(\gamma_{ij}), \#L(p) \geq 2; \\ 0, \quad\quad\quad\quad\quad \#L(P) = 1, \end{cases} \tag{5}$$

*where* $f : [0,1] \to [0,1]$ *monotone increasing,* $\gamma_{ij} = |\gamma_i - \gamma_j| i, j = 1, 2, \cdots, \#L(P)$, $\gamma_i, \gamma_j \in g(L)$, $f(0) = 0, f(1) = 1$. *the entropy measure describes the degree of uncertainty contained in language terms and their corresponding probability information.*

2.　Probabilistic linguistic interaction-entropy measure

If the deviation between the decision information given by expert $E_i$ and other experts is greater, the larger the measurement value of interaction entropy is, indicating that the decision information given by expert $E_i$ is less reliable. In this case, the weight given by expert $E_i$ should be smaller [15].

$$\omega'_i = \frac{\sum_{h=1, h \neq i}^{k} \sum_{j=1}^{n} (1 - IE(L_{ij}(p), L_{hj}(p)))}{\sum_{i=1}^{k} \sum_{h=1, h \neq i}^{k} \sum_{j=1}^{n} (1 - IE(L_{ij}(p), L_{hj}(p)))} (i = 1, 2, \cdots, m) \tag{6}$$

where $IE(L_{ij}(p), L_{hj}(p))$ is the interaction-entropy measure of PLTS $L_{ij}(p), L_{hj}(p)$. It is defined as follows.

**Theorem 4.** *Ref.* [15] $L_1(p) = \{L_1^k(p_1^k) | k = 1, 2, \cdots, \#L(P)\}$ *and* $L_2(p) = \{L_2^k(p_2^k) | k = 1, 2, \cdots, \#L(P)\}$ *are two PLTSs. The interaction-entropy measure of PLTS is:*

$$IE(L_1(p), L_2(p)) = \frac{1}{2} (RE(L_1(p), L_2(p)) + RE(L_2(p), L_1(p))) \tag{7}$$

$$RE(L_1(p), L_2(p)) = \sum_{k=1}^{\#L(p)} p_1^k (g(L_1^k) \log \frac{g(L_1^k)}{g(L_2^k)} + (1 - g(L_1^k)) \log(\frac{1 - g(L_1^k)}{1 - g(L_2^k)})) \tag{8}$$

Through the above entropy measure, we can obtain the final objective weight of expert $E_i$:

$$\omega_i = \frac{\omega_i \omega'_i}{\sum_{i=1}^{k} \omega_i \omega'_i} (i = 1, 2, \cdots, m) \tag{9}$$

**Theorem 5.** *Ref. [15] An PLTS $L(p) = \left\{ L^k(p^k) | L^k \in S.p^k \geq 0.k = 1, 2, \cdots, \#L(p) \right\}$ and a hesitation fuzzy set $h_\gamma = \left\{ \gamma^k | \gamma \in [0, 1] \right\}$, the membership $\gamma^k$ $inh_\gamma$ and the linguistic term $L^k$ in $L(p)$ can be transformed into each other by the equivalent functions $g$ and $g^{-1}$, which are defined as:*

$$g : [0, 2\tau] \rightarrow [0, 1], g(L^k) = \frac{ind(L^k)}{2\tau} = \gamma^k$$
$$g^{-1} : [0, 1] \rightarrow [0, 2\tau], g^{-1}(\gamma^k) = s_{2\tau\gamma^k} = L^k \tag{10}$$

3. The final weight of experts:

$$\omega = (\omega(E_i) + \omega_i)/2 \tag{11}$$

**Step 4. Transform PLTS into triangular fuzzy number**

**Theorem 6.** *Ref. [42] A three tuple $A = (a, b, c)$ is called a TFN if it satisfies:*

$$\mu_A(x) = \begin{cases} 0, x \prec a \\ \frac{x-a}{b-a}, a \leq x \leq b \\ \frac{c-x}{c-b}, b \leq x \leq c \\ 0, x \succ c \end{cases} \tag{12}$$

**Theorem 7.** *Ref. [43] For any set of language terms $s_k(k = i, i+1, \cdots, j)$, we can describe it by a TFN $A_k = (a_k^L, a_k^M, a_k^R)$, where:*

(1) If $-\tau \prec k \prec \tau$, then $a_k^L = \frac{\tau+k-1}{2\tau}, a_k^M = \frac{\tau+k}{2\tau}, a_k^R = \frac{\tau+k+1}{2\tau}$
(2) If $k = -\tau$, then $a_k^L = a_k^M = 0, a_k^R = \frac{1}{2\tau}$
(3) If $k = \tau$, then $a_k^L = \frac{2\tau-1}{2\tau}, a_k^M = a_k^R = 1$

**Example 2.** *$\tau = 2$, TFN used to represent the language term is $s_{-2} = (0, 0, 0.25)$, $s_{-1} = (0, 0.25, 0.5)$, $s_0 = (0.25, 0.5, 0.75)$, $s_1 = (0.5, 0.75, 1)$, $s_2 = (0.75, 1, 1)$.*

According to the above definition, we can obtain the numerical set corresponding to TFN of all language terms in PLTS $T = \left\{ a_i^L, a_i^M, a_{i+1}^L, a_i^R, a_{i+1}^M, a_{i+2}^L, a_{i+1}^R, \cdots, a_j^L, a_{j-1}^R, a_j^M, a_j^R \right\}$, and $a_{k-1}^R = a_k^M = a_{k+1}^L, k = 1, \cdots j-1$, then $T = \left\{ a_i^L, a_i^M, a_{i+1}^M, \cdots, a_j^M, a_j^R \right\}$.

To transform PLTS into TFN, PLTS information must be fully utilized. In this article, we used the GWOWA operator to aggregate information. PLTS have two key elements: language terms and their probabilities. The information of language terms can be utilized by transforming them into TFNs. In addition, their probability information can be embedded using the importance-weighted vector of the GWOWA operator, as defined and described below:

Since the subscripts of language terms may not be continuous, we add some elements with a probability value of 0 in PLTS to make the subscripts of language terms continuous.

**Example 3.** *$L(p) = \{(s_{-3}, 0.2), (s_0, 0.8)\}$ is a PLTS, we add some elements and rewrite it as $L(p) = \{(s_{-3}, 0.2), (s_{-2}, 0), (s_{-1}, 0), (s_0, 0.8)\}$.*

**Theorem 8.** *Ref. [20] $P = (p_1, p_2, \cdots, p_n)$ is the weight information of $a_1, a_2, \cdots, a_n$, $p_i \in [0, 1], \sum_{i=1}^n p_i = 1$, then, $f_{GWOWA}^{P,W} : R^n \rightarrow R$, the corresponding weight information $W = (\omega_1, \omega_2, \cdots, \omega_n), \omega_i \in [0, 1], \sum_{i=1}^n \omega_i = 1$, the definition of the GWOWA operator is:*

$$f_{GWOWA}^{P,W}(a_1, a_2, \cdots, a_n) = \left(\sum_{i=1}^{n} v_i b_i^{\lambda}\right)^{\frac{1}{\lambda}} \tag{13}$$

where $\lambda \in (0, +\infty)$, $b_i$ is the $i$th largest element in $a_1, a_2, \cdots, a_n$, the weight $v_i$ is calculated as follows:

$$v_i = \omega^*\left(\sum_{j=1}^{i} p_\sigma(j)\right) - \omega^*\left(\sum_{j=1}^{i-1} p_\sigma(j)\right) \tag{14}$$

where $\omega *$ is a monotonically increasing function. It can be written as:

$$\omega^*(x) = \sum_{k=1}^{i-1} \omega_k + \omega_i(nx - (i-1)), \frac{i-1}{n} \le x \le \frac{1}{n} \tag{15}$$

For convenience, $P = (p_1, p_2, \cdots, p_n)$, $W = (\omega_1, \omega_2, \cdots \omega_n)$, $V = (v_1, v_2, \cdots, v_n)$ are called the importance weighting vector, position weighting vector, and the comprehensive weighting vector, respectively. Different methods exist to compute position weighting vector $W = (\omega_1, \omega_2, \cdots \omega_n)$. Yager [44] proposed the attitudinal character as $AC(W) = \sum_{j=1}^{n} \frac{n-j}{n-1}\omega_j$, which expresses the attitude of the DM for giving more weight to higher or lower values. The weights can be calculated using the following mathematical programming:

$$
\begin{aligned}
&Max \sum_{j=1}^{n} \omega_j \ln(\omega_j) \\
&s.t. \sum_{j=1}^{n} \frac{n-i}{n-1}\omega_j = \alpha \\
&\quad \sum_{j=1}^{n} \omega_j = 1 \\
&\quad 0 \le \omega_j \le 1 (j = 1, 2, \cdots, n)
\end{aligned}
\tag{16}
$$

where attitudinal character $\alpha$ should be given by DMs.

For $L(p) = \{s_i(p_i), s_{i+1}(p_{i+1}), \cdots, s_j(p_j) | s_k \in S, k \in \{i, i+1, \cdots, j\}, p_i \ge 0, \sum_{k=i}^{j} p_k = 1\}$, we transform the PLTS into a TFN $A = (a, b, c)$, where:

$$a = \min\{a_i^L, a_i^M, a_{i+1}^L, a_i^R, a_{i+1}^M, a_{i+2}^L, a_{i+1}^R, \cdots, a_j^L, a_{j-1}^R, a_j^M, a_j^R\} \tag{17}$$

$$c = \max\{a_i^L, a_i^M, a_{i+1}^L, a_i^R, a_{i+1}^M, a_{i+2}^L, a_{i+1}^R, \cdots, a_j^L, a_{j-1}^R, a_j^M, a_j^R\} \tag{18}$$

The value of $b$ is determined according to the following rules:

(1) If $j = \tau$, then $a_\tau^M = a_\tau^R$, $T = \{a_i^L, a_i^M, a_{i+1}^M, \cdots, a_\tau^M, a_\tau^R\}$. Using the GWOWA operator, parameter $b$ of the TFN can be calculated as $b = f_{GWOWA}^{P,M}(a_i^M, a_{i+1}^M, \cdots, a_\tau^M)$, where the elements of the importance weighting vector $P = (p_i, p_{i+1}, \cdots, p_\tau)$ correspond to the probability values in $L(p)$. PLTS $L(p)$ can be translated into TFN:

$$A = (a_i^L, f_{GWOWA}^{P,M}(a_i^M, a_{i+1}^M, \cdots, a_\tau^M), a_\tau^M) \tag{19}$$

(2) If $j = -\tau$, then $a_{-\tau}^L = a_{-\tau}^M$, $T = \{a_{-\tau}^L, a_{-\tau}^M, a_{-\tau+1}^M, \cdots, a_j^M, a_j^R\}$, we can easily obtain:

$$a = \min\{a_{-\tau}^M, a_{-\tau}^M, a_{-\tau+1}^M, \cdots, a_j^M, a_j^R\} = a_{-\tau}^M \tag{20}$$

$$c = \max\{a_{-\tau}^L, a_{-\tau}^M, a_{-\tau+1}^M, \cdots, a_j^M, a_j^R\} = a_j^R \tag{21}$$

$$b = f_{GWOWA}^{P,M}(a_{-\tau}^M, a_{-\tau+1}^M, \cdots, a_j^M) \tag{22}$$

PLTS $L(p)$ can be translated into TFN:

$$A = (a^M_{-\tau}, f^{P,M}_{GWOWA}(a^M_{-\tau}, a^M_{-\tau+1}, \cdots, a^M_j), a^R_j) \tag{23}$$

(3) If $i \succ -\tau, j \prec \tau$, we can obtain $T = \left\{ a^L_i, a^M_i, a^M_1, \cdots, a^M_j, a^R_j \right\}$, we can easily obtain:

$$a = \min\{a^L_i, a^M_i, a^L_{i+1}, a^R_i, a^M_{i+1}, a^L_{i+2}, a^R_{i+1}, \cdots, a^L_j, a^R_{j-1}, a^M_j, a^R_j\} = a^L_i \tag{24}$$

$$c = \max\{a^L_i, a^M_i, a^L_{i+1}, a^R_i, a^M_{i+1}, a^L_{i+2}, a^R_{i+1}, \cdots, a^L_j, a^R_{j-1}, a^M_j, a^R_j\} = a^R_j \tag{25}$$

Using the GWOWA operator, the value of $b$ in TFN can be calculated $b = f^{P,M}_{GWOWA}(a^M_i, a^M_{i+1}, \cdots, a^M_j)$, PLTS $L(p)$ can be translated into TFN:

$$A = (a^L_i, f^{P,M}_{GWOWA}(a^M_i, a^M_{i+1}, \cdots, a^M_j), a^R_j) \tag{26}$$

**Example 4.** $S = \{s_{-3}, s_{-2}, s_{-1}, s_0, s_1, s_2, s_3\}$, *we take position weighting vector* $W = (0.44, 0.23, 0.33)$, *and* $\lambda = 1$ *[25]. PLTS* $L(p) = \{(s_1, 1/3), (s_2, 1/3), (s_3, 1/3)\}$ *can be transformed into TFN* $A = (0.5, 0.85, 1)$.

**Step 5. Aggregate expert evaluation information**
To realize the aggregation of expert evaluation information, a fuzzy induced ordered weighted harmonic averaging (FIOWHA) operator based on TFN was adopted in this paper, which is defined as follows:

**Theorem 9.** *Ref. [14]* $a_1, a_2, \cdots, a_n$ *is a set of TFNs to be aggregated. The definition of the FIOWHA operator is:*

$$FIOWHA_\omega((\mu_1, \gamma_1), (\mu_2, \gamma_2), \cdots, (\mu_n, \gamma_n)) = \frac{1}{\sum\limits_{j=1}^{n} \frac{\omega_j}{g_j}} \tag{27}$$

where $\gamma_j = [\gamma^L_j, \gamma^M_j, \gamma^U_j], \omega = (\omega_1, \omega_2, \cdots, \omega_n)^T$ is a weight vector associated with the FIOWHA operator that satisfies $\omega_j \in [0, 1]$, and $\sum\limits_{j=1}^{n} \omega_j = 1$. $g_j$ is the second vector $\gamma_i$ in $(\mu_i, \gamma_i)$ of the $i$th largest element in $\mu_i(i = 1, 2, \cdots, n)$. The first vector $\mu_i$ in $(\mu_i, \gamma_i)$ is called the order-induced vector.

**Step 6. Aggregating TFN of indicator based on FSE method**
The total risk level of the project is obtained by aggregating the triangle fuzzy number of each risk index with the fuzzy synthetic evaluation method. The process of the FSE method is classified into three stages. Firstly, a first-order evaluation vector composed of TFNs of criteria in each group is established $R_{ci}$:

$$R_{ci} = (h_{ci1}, \cdots, h_{cij})^T \tag{28}$$

Secondly, the secondary evaluation vector $R_c$ containing each group of triangular fuzzy numbers is obtained using fuzzy synthesis operation:

$$h_{ci} = W_{ci} \cdot R_{ci} = (\omega_{ci1}, \cdots, \omega_{cij}) \cdot (h_{ci1}, \cdots, h_{cij})^T \tag{29}$$

$$R_c = (h_{c1}, \cdots, h_{ci}) \tag{30}$$

where $\omega_{cij}$ is the weight of each risk factor within every group.

Thirdly, the overall risk level of the project represented by TNFs is calculated as:

$$R = W_i \cdot R_c = (\omega_{c1}, \cdots, \omega_{ci}) \cdot (h_{ci}, \cdots, h_{ci})^T = (\gamma^L, \gamma^M, \gamma^U) \tag{31}$$

where $\omega_{ci}$ denotes the weight of each risk factor group.

**Step 7. Defuzzification process**

TFN can be used to represent the risk level of hybrid offshore wind–solar PV power generation projects. The similarity degree is introduced for the defuzzification process in this paper to get a more intuitive and easy-to-understand result. The similarity degree between two TFNs can be calculated as [14]:

$$Sd(\alpha, \beta) = 1 - \frac{|\alpha^L - \beta^L| + |\alpha^M - \beta^M| + |\alpha^U - \beta^U|}{3} \tag{32}$$

where $\alpha = (\alpha^L, \alpha^M, \alpha^U), \beta = (\beta^L, \beta^M, \beta^U)$ are TFNs, and $Sd(\alpha, \beta)$ represents the similarity degree between $\alpha$ and $\beta$. Thus, which risk level the evaluation result is closer to can be obtained using the principle of maximum similarity. Finally, the risk assessment framework is shown in Figure 3.

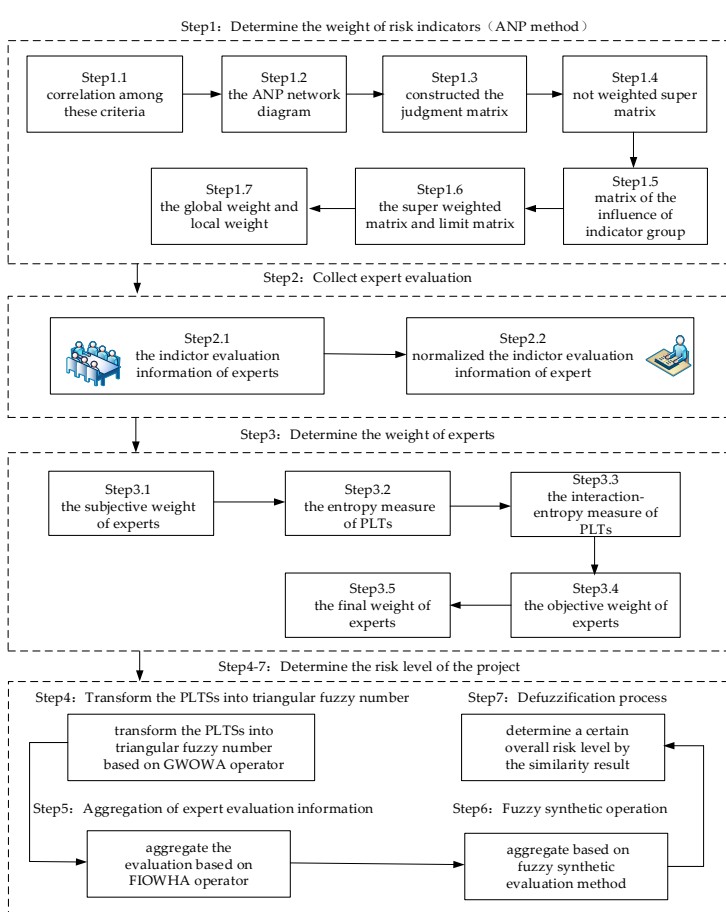

**Figure 3.** The risk assessment framework for hybrid offshore wind–solar PV power.

## 5. Case Study

In order to demonstrate the rationality and usability of the proposed risk assessment framework for hybrid offshore wind–solar PV power generation projects, a case study is presented in this part.

### 5.1. Problem Description

B city plans to implement the construction of a hybrid offshore wind–solar PV power generation project. Before the implementation of the project, it plans to conduct a risk assessment on the project. Three experts $E = \{e_i | i = 1, 2, 3\}$ are invited to evaluate the risk indicators of the project and express them in the form of PLTSs.

### 5.2. Determination of the Weight of Risk Indicators

The analytic network process (ANP) method was used to determine the weight of risk indicators. First, the ANP structure model was constructed, and the ANP structure model was followed by decision-making objectives, decision-making criteria, and indicators from top to bottom. Invited related scholars, who have rich experiences in offshore energy projects, set up a committee of experts. The expert committee discussed the interaction relationship between various risk indicators and established an ANP network diagram; Figure 4 shows the ANP network diagram. The straight arrows indicate that the indicators in the two indicator groups have an influence relationship, and the curved arrows indicate that there is an interaction relationship between the indicators in the indicator group. At the same time, the expert committee made a pairwise comparison of indicators on 1–9 scales under different criteria, constructed a judgment matrix, and conducted a consistency test to calculate its weight vector, taking C11 criteria as an example (Tables 2–6). After computing the judgment matrix, the unweighted super matrix was generated by the software. In addition, we compared the relationships between various indicator groups, and input the compare matrix into the software and obtained the influence matrix of the indicator group (shown in Table 7). The super weighted matrix and limit matrix were obtained using software calculation, and the global weight and local weight of the indicator were obtained (Table 8, Figure 5).

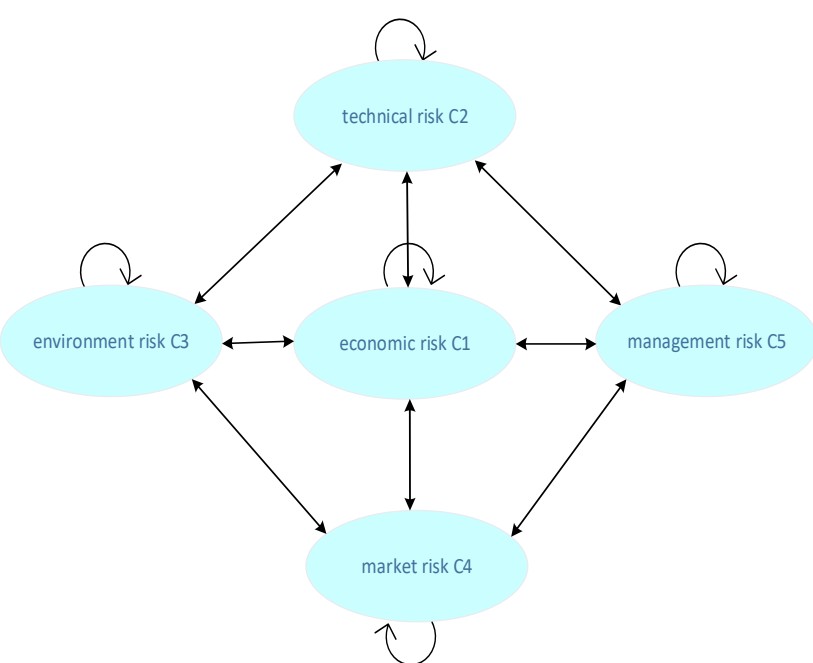

**Figure 4.** The ANP network diagram.

**Table 2.** The judgement matrix of C1 under the C11 criterion.

| C11 | C12 | C13 | C14 | Weight |
|---|---|---|---|---|
| C12 | 1 | 4 | 2 | 0.54 |
| C13 | 1/4 | 1 | 1/3 | 0.13 |
| C14 | 1/2 | 3 | 1 | 0.34 |

CR = 0 < 0.1, consistency test passed.

**Table 3.** The judgement matrix of C2 under the C11 criterion.

| C11 | C21 | C22 | C23 | C24 | C25 | C26 | Weight |
|-----|-----|-----|-----|-----|-----|-----|--------|
| C21 | 1 | 1 | 3 | 2 | 4 | 2 | 0.24 |
| C22 | 1 | 1 | 4 | 3 | 6 | 2 | 0.31 |
| C23 | 1/3 | 1/4 | 1 | 1/2 | 4 | 1/2 | 0.12 |
| C24 | 1/2 | 1/3 | 2 | 1 | 3 | 1 | 0.14 |
| C25 | 1/4 | 1/6 | 1/4 | 1/3 | 1 | 1/3 | 0.04 |
| C26 | 1/2 | 1/2 | 2 | 1 | 3 | 1 | 0.15 |

CR = 0 < 0.1, consistency test passed.

**Table 4.** The judgement matrix of C3 under the C11 criterion.

| C11 | C31 | C32 | C33 | Weight |
|-----|-----|-----|-----|--------|
| C31 | 1 | 1 | 3 | 0.43 |
| C32 | 1 | 1 | 3 | 0.43 |
| C33 | 1/3 | 1/3 | 1 | 0.14 |

CR = 0 < 0.1, consistency test passed.

**Table 5.** The judgement matrix of C4 under the C11 criterion.

| C11 | C41 | C42 | C43 | C44 | Weight |
|-----|-----|-----|-----|-----|--------|
| C41 | 1 | 1 | 4 | 1/2 | 0.3 |
| C42 | 1 | 1 | 4 | 1/2 | 0.3 |
| C43 | 1/4 | 1/4 | 1 | 1 | 0.12 |
| C44 | 2 | 2 | 1 | 1 | 0.28 |

CR = 0 < 0.1, consistency test passed.

**Table 6.** The judgement matrix of C5 under the C11 criterion.

| C11 | C51 | C52 | Weight |
|-----|-----|-----|--------|
| C51 | 1 | 1 | 0.5 |
| C52 | 1 | 1 | 0.5 |

CR = 0 < 0.1, consistency test passed.

**Table 7.** The influence matrix of the indicator group.

| Indicator | C1 | C2 | C3 | C4 | C5 |
|-----------|------|------|------|------|------|
| C1 | 0.169 | 0.161 | 0.293 | 0.303 | 0.21 |
| C2 | 0.333 | 0.267 | 0.178 | 0.197 | 0.193 |
| C3 | 0.264 | 0.37 | 0.233 | 0.247 | 0.333 |
| C4 | 0.16 | 0.134 | 0.109 | 0.13 | 0.134 |
| C5 | 0.074 | 0.068 | 0.26 | 0.123 | 0.13 |

**Table 8.** The global weight and local weight of the indicator.

| Indicator | Indicator | The Local Weight | The Global Weight |
|-----------|-----------|------------------|-------------------|
| Economic risk C1 (0.3145) | C11 | 0.2031 | 0.0638 |
| | C12 | 0.3352 | 0.1054 |
| | C13 | 0.0978 | 0.0308 |
| | C14 | 0.3639 | 0.1145 |
| Technical risk C2 (0.1903) | C21 | 0.2039 | 0.0388 |
| | C22 | 0.2142 | 0.0408 |
| | C23 | 0.1302 | 0.0248 |

**Table 8.** *Cont.*

| Indicator | Indicator | The Local Weight | The Global Weight |
|---|---|---|---|
| | C24 | 0.1703 | 0.0324 |
| | C25 | 0.0948 | 0.018 |
| | C26 | 0.1866 | 0.0355 |
| Environment risk C3 (0.2501) | C31 | 0.3261 | 0.0816 |
| | C32 | 0.3142 | 0.0786 |
| | C33 | 0.3597 | 0.0899 |
| Market risk C4 (0.1519) | C41 | 0.2772 | 0.0421 |
| | C42 | 0.2830 | 0.043 |
| | C43 | 0.1271 | 0.0193 |
| | C44 | 0.3127 | 0.0475 |
| Management risk C5 (0.0932) | C51 | 0.6476 | 0.0604 |
| | C52 | 0.3524 | 0.0328 |

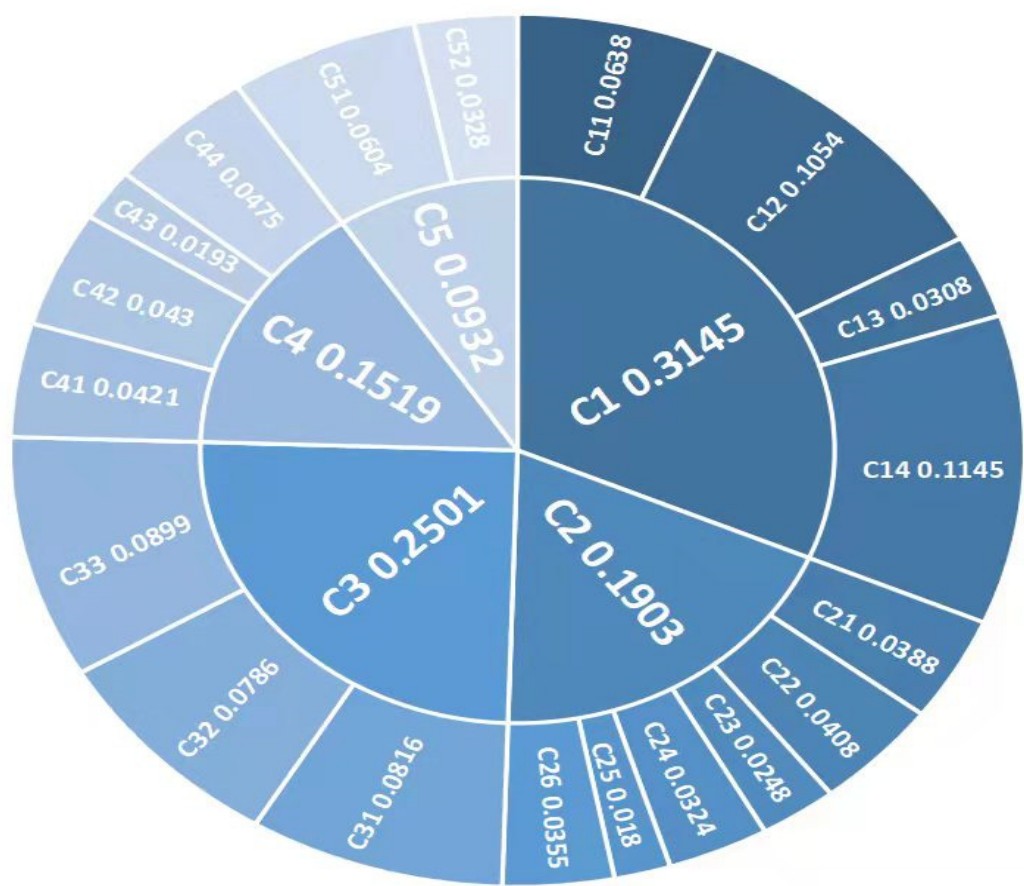

**Figure 5.** The weight of indicators.

### 5.3. Collect Expert Evaluation

We set up a committee of experts, $E = \{e_i | i = 1, 2, 3\}$. Probabilistic language term sets were used to evaluate the risk assessment indicator system of hybrid offshore wind–solar PV power generation projects constructed in the third part. In this paper, $S = \{s_0 : VL, s_1 : L, s_2 : SL, s_3 : M, s_4 : SH, s_5 : H, s_6 : VH\}$, which is defined as a seven-scale language term set, where very low (VL), low (L), low (L), slightly low (SL), medium

(M), slightly high (SH), high (H), and very high (VH). Table 9 shows the original evaluation data and Table 10 shows the normalized indictor evaluation information.

**Table 9.** The indictor evaluation information of expert $E_i$.

| Indicator | $E_1$ | $E_2$ | $E_3$ |
|---|---|---|---|
| C11 | $\{s_4(0.7), s_5(0.3)\}$ | $\{s_3(0.4), s_4(0.6)\}$ | $\{s_4(1)\}$ |
| C12 | $\{s_6(1)\}$ | $\{s_5(0.5), s_6(0.5)\}$ | $\{s_4(0.3), s_5(0.7)\}$ |
| C13 | $\{s_3(0.6), s_4(0.4)\}$ | $\{s_3(1)\}$ | $\{s_2(0.5), s_3(0.5)\}$ |
| C14 | $\{s_5(0.7), s_6(0.3)\}$ | $\{s_5(0.55), s_6(0.45)\}$ | $\{s_5(0.9)\}$ |
| C21 | $\{s_5(0.9)\}$ | $\{s_4(0.5), s_5(0.5)\}$ | $\{s_4(0.43), s_5(0.57)\}$ |
| C22 | $\{s_4(0.7), s_5(0.3)\}$ | $\{s_4(0.4), s_5(0.6)\}$ | $\{s_5(0.9), s_6(0.1)\}$ |
| C23 | $\{s_4(0.75), s_5(0.25)\}$ | $\{s_4(1)\}$ | $\{s_3(0.1), s_4(0.9)\}$ |
| C24 | $\{s_4(0.4), s_5(0.6)\}$ | $\{s_4(0.9)\}$ | $\{s_4(0.5), s_5(0.5)\}$ |
| C25 | $\{s_1(0.1), s_2(0.9)\}$ | $\{s_1(0.4), s_2(0.6)\}$ | $\{s_2(0.9)\}$ |
| C26 | $\{s_4(0.5), s_5(0.5)\}$ | $\{s_4(1)\}$ | $\{s_4(0.5), s_5(0.5)\}$ |
| C31 | $\{s_4(0.4), s_5(0.6)\}$ | $\{s_5(1)\}$ | $\{s_5(0.55), s_6(0.45)\}$ |
| C32 | $\{s_4(0.5), s_5(0.5)\}$ | $\{s_5(0.7), s_6(0.3)\}$ | $\{s_4(0.7), s_5(0.3)\}$ |
| C33 | $\{s_4(0.7), s_5(0.3)\}$ | $\{s_5(0.8)\}$ | $\{s_4(0.5), s_5(0.5)\}$ |
| C41 | $\{s_4(0.5), s_5(0.5)\}$ | $\{s_4(0.7), s_5(0.3)\}$ | $\{s_4(0.9), s_5(0.1)\}$ |
| C42 | $\{s_4(0.57), s_5(0.43)\}$ | $\{s_5(0.8)\}$ | $\{s_4(0.9), s_5(0.1)\}$ |
| C43 | $\{s_2(0.2), s_3(0.8)\}$ | $\{s_3(0.5), s_4(0.5)\}$ | $\{s_2(0.2), s_3(0.8)\}$ |
| C44 | $\{s_2(0.6), s_3(0.3)\}$ | $\{s_2(0.6), s_3(0.4)\}$ | $\{s_3(0.9)\}$ |
| C51 | $\{s_4(0.6), s_5(0.4)\}$ | $\{s_3(0.8)\}$ | $\{s_3(0.5), s_4(0.5)\}$ |
| C52 | $\{s_3(0.7), s_4(0.3)\}$ | $\{s_2(0.2), s_3(0.8)\}$ | $\{s_3(0.5), s_4(0.5)\}$ |

**Table 10.** Normalized indictor evaluation information of expert $E_i$.

| Indicator | $E_1$ | $E_2$ | $E_3$ |
|---|---|---|---|
| C11 | $\{s_4(0.7), s_5(0.3)\}$ | $\{s_3(0.4), s_4(0.6)\}$ | $\{s_4(0), s_4(1)\}$ |
| C12 | $\{s_6(0), s_6(1)\}$ | $\{s_5(0.5), s_6(0.5)\}$ | $\{s_4(0.3), s_5(0.7)\}$ |
| C13 | $\{s_3(0.6), s_4(0.4)\}$ | $\{s_3(0), s_3(1)\}$ | $\{s_2(0.5), s_3(0.5)\}$ |
| C14 | $\{s_5(0.7), s_6(0.3)\}$ | $\{s_5(0.55), s_6(0.45)\}$ | $\{s_5(0), s_5(1)\}$ |
| C21 | $\{s_5(0), s_5(1)\}$ | $\{s_4(0.5), s_5(0.5)\}$ | $\{s_4(0.43), s_5(0.57)\}$ |
| C22 | $\{s_4(0.7), s_5(0.3)\}$ | $\{s_4(0.4), s_5(0.6)\}$ | $\{s_5(0.9), s_6(0.1)\}$ |
| C23 | $\{s_4(0.75), s_5(0.25)\}$ | $\{s_4(0), s_4(1)\}$ | $\{s_3(0.1), s_4(0.9)\}$ |
| C24 | $\{s_4(0.4), s_5(0.6)\}$ | $\{s_4(0), s_4(1)\}$ | $\{s_4(0.5), s_5(0.5)\}$ |
| C25 | $\{s_1(0.1), s_2(0.9)\}$ | $\{s_1(0.4), s_2(0.6)\}$ | $\{s_2(0), s_2(1)\}$ |
| C26 | $\{s_4(0.5), s_5(0.5)\}$ | $\{s_4(0), s_4(1)\}$ | $\{s_4(0.5), s_5(0.5)\}$ |
| C31 | $\{s_4(0.4), s_5(0.6)\}$ | $\{s_5(0), s_5(1)\}$ | $\{s_5(0.55), s_6(0.45)\}$ |
| C32 | $\{s_4(0.5), s_5(0.5)\}$ | $\{s_5(0.7), s_6(0.3)\}$ | $\{s_4(0.7), s_5(0.3)\}$ |
| C33 | $\{s_4(0.7), s_5(0.3)\}$ | $\{s_5(0), s_5(1)\}$ | $\{s_4(0.5), s_5(0.5)\}$ |
| C41 | $\{s_4(0.5), s_5(0.5)\}$ | $\{s_4(0.7), s_5(0.3)\}$ | $\{s_4(0.9), s_5(0.1)\}$ |
| C42 | $\{s_4(0.57), s_5(0.43)\}$ | $\{s_5(0), s_5(1)\}$ | $\{s_4(0.9), s_5(0.1)\}$ |
| C43 | $\{s_2(0.2), s_3(0.8)\}$ | $\{s_3(0.5), s_4(0.5)\}$ | $\{s_2(0.2), s_3(0.8)\}$ |
| C44 | $\{s_2(0.67), s_3(0.33)\}$ | $\{s_2(0.6), s_3(0.4)\}$ | $\{s_3(0), s_3(1)\}$ |
| C51 | $\{s_4(0.6), s_5(0.4)\}$ | $\{s_3(0), s_3(1)\}$ | $\{s_3(0.5), s_4(0.5)\}$ |
| C52 | $\{s_3(0.7), s_4(0.3)\}$ | $\{s_2(0.2), s_3(0.8)\}$ | $\{s_3(0.5), s_4(0.5)\}$ |

## 5.4. Determine the Weight of Experts

The weight of experts adopted the method of combining subjective and objective weights. The calculation of subjective weight of experts is as shown in Part 4. The information of the expert committee is shown in the following Table 11.

**Table 11.** The information of the expert committee.

| Expert | Position | Project | Education | Working | Weight |
|--------|----------|---------|-----------|---------|--------|
| $E_1$ | 4 | 3 | 3 | 3 | 0.433 |
| $E_2$ | 2 | 1 | 2 | 3 | 0.267 |
| $E_3$ | 2 | 2 | 2 | 3 | 0.3 |

$\omega(E_i) = (0.433, 0.267, 0.3)$ is the subjective weight of experts. According to the equation of step 3 in part 4, assume $f(x) = x^r, r = 1$. The expert probabilistic language entropy measure is calculated as $\varpi_i = (0.3238, 0.3396, 0.3365)$, and the probabilistic language inter-entropy measure is calculated as $\omega'_i = (0.3332, 0.3327, 0.3341)$. The final weight of experts is $\omega_i = (0.3784, 0.303, 0.3186)$.

### 5.5. Transform the PLTSs into Triangular Fuzzy Number

Before transforming the PLTSs into TFN based on the GWOWA operator, first $S = \{s_0 : VL, s_1 : L, s_2 : SL, s_3 : M, s_4 : SH, s_5 : H, s_6 : VH\}$ was transformed into the language term set $S = \{s_{-3} : VL, s_{-2} : L, s_{-1} : SL, s_0 : M, s_1 : SH, s_2 : H, s_3 : VH\}$ with $\tau = 3$. The position weight $\omega = (0.47, 0.53)$ was assumed, and $\lambda = 1$ [25]. The value of triangular fuzzy number is shown in Figure 6. The results of triangular fuzzy numbers are shown in Table 12.

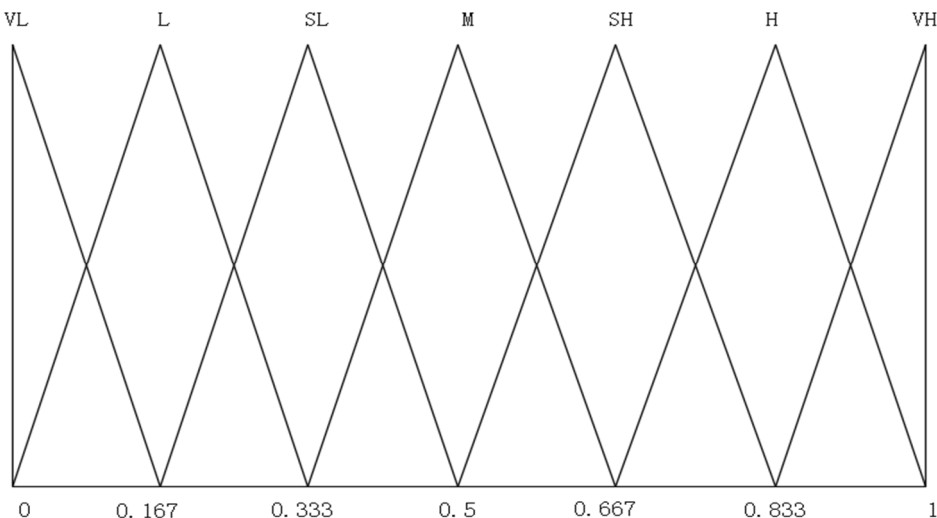

**Figure 6.** The value of the triangular fuzzy number.

**Table 12.** Transformation of the PLTSs into triangular fuzzy numbers.

| Indicator | $E_1$ | $E_2$ | $E_3$ |
|-----------|-------|-------|-------|
| C11 | $(0.5, 0.776, 1)$ | $(0.333, 0.565, 0.833)$ | $(0.5, 0.667, 0.833)$ |
| C12 | $(0.833, 1, 1)$ | $(0.667, 0.91, 1)$ | $(0.5, 0.715, 1)$ |
| C13 | $(0.333, 0.595, 0.833)$ | $(0.333, 0.5, 0.667)$ | $(0167, 0.378, 0.667)$ |
| C14 | $(0.667, 0.942, 1)$ | $(0.667, 0.918, 1)$ | $(0.667, 0.833, 1)$ |
| C21 | $(0.667, 0.833, 1)$ | $(0.5, 0.745, 1)$ | $(0.5, 0.738, 1)$ |
| C22 | $(0.5, 0.776, 1)$ | $(0.5, 0.731, 1)$ | $(0.667, 0.954, 1)$ |
| C23 | $(0.5, 0.783, 1)$ | $(0.5, 0.667, 0.833)$ | $(0.333, 0.515, 0.833)$ |
| C24 | $(0.5, 0.731, 1)$ | $(0.5, 0.667, 0.833)$ | $(0.5, 0.745, 1)$ |
| C25 | $(0, 0.184, 0.5)$ | $(0, 0.231, 0.5)$ | $(0.167, 0.333, 0.5)$ |
| C26 | $(0.5, 0.745, 1)$ | $(0.5, 0.667, 0.833)$ | $(0.5, 0.731, 1)$ |
| C31 | $(0.5, 0.731, 1)$ | $(0.667, 0.833, 1)$ | $(0.5, 0.753, 1)$ |
| C32 | $(0.5, 0.745, 1)$ | $(0.667, 0.942, 1)$ | $(0.5, 0.776, 1)$ |
| C33 | $(0.5, 0.776, 1)$ | $(0.667, 0.833, 1)$ | $(0.5, 0.745, 1)$ |
| C41 | $(0.5, 0.745, 1)$ | $(0.5, 0.714, 1)$ | $(0.5, 0.684, 1)$ |

**Table 12.** *Cont.*

| Indicator | $E_1$ | $E_2$ | $E_3$ |
|---|---|---|---|
| C42 | $(0.5, 0.756, 1)$ | $(0.667, 0.833, 1)$ | $(0.5, 0.684, 1)$ |
| C43 | $(0.5, 0.756, 1)$ | $(0.333, 0.58, 0.833)$ | $(0.167, 0.362, 0.667)$ |
| C44 | $(0.167, 0.437, 0.667)$ | $(0.167, 0.425, 0.667)$ | $(0.333, 0.5, 0.667)$ |
| C51 | $(0.5, 0.76, 1)$ | $(0.333, 0.5, 0.667)$ | $(0.333, 0.58, 0.833)$ |
| C52 | $(0.333, 0.612, 0.833)$ | $(0.167, 0.362, 0.667)$ | $(0.333, 0.58, 0.833)$ |

*5.6. Aggregation of Expert Evaluation Information*

In this paper, the fuzzy induced ordered weighted harmonic average (FIOWHA) operator of TFN was used for aggregation, and the aggregation results are shown in Table 13.

**Table 13.** The result of aggregation of triangular fuzzy numbers.

| Indicator | $E_1$ | $E_2$ | $E_3$ | Result |
|---|---|---|---|---|
| C11 | $(0.5, 0.776, 1)$ | $(0.333, 0.565, 0.833)$ | $(0.5, 0.667, 0.833)$ | $(0.449, 0.677, 0.896)$ |
| C12 | $(0.833, 1, 1)$ | $(0.667, 0.91, 1)$ | $(0.5, 0.715, 1)$ | $(0.677, 0.882, 1)$ |
| C13 | $(0.333, 0.595, 0.833)$ | $(0.333, 0.5, 0.667)$ | $(0.167, 0.378, 0.667)$ | $(0.28, 0.497, 0.73)$ |
| C14 | $(0.667, 0.942, 1)$ | $(0.667, 0.918, 1)$ | $(0.667, 0.833, 1)$ | $(0.667, 0.9, 1)$ |
| C21 | $(0.667, 0.833, 1)$ | $(0.5, 0.745, 1)$ | $(0.5, 0.738, 1)$ | $(0.563, 0.776, 1)$ |
| C22 | $(0.5, 0.776, 1)$ | $(0.5, 0.731, 1)$ | $(0.667, 0.954, 1)$ | $(0.553, 0.819, 1)$ |
| C23 | $(0.5, 0.783, 1)$ | $(0.5, 0.667, 0.833)$ | $(0.333, 0.515, 0.833)$ | $(0.447, 0.662, 0.896)$ |
| C24 | $(0.5, 0.731, 1)$ | $(0.5, 0.667, 0.833)$ | $(0.5, 0.745, 1)$ | $(0.5, 0.716, 0.949)$ |
| C25 | $(0, 0.184, 0.5)$ | $(0, 0.231, 0.5)$ | $(0.167, 0.333, 0.5)$ | $(0.053, 0.246, 0.5)$ |
| C26 | $(0.5, 0.745, 1)$ | $(0.5, 0.667, 0.833)$ | $(0.5, 0.731, 1)$ | $(0.5, 0.717, 0.949)$ |
| C31 | $(0.5, 0.731, 1)$ | $(0.667, 0.833, 1)$ | $(0.5, 0.753, 1)$ | $(0.551, 0.769, 1)$ |
| C32 | $(0.5, 0.745, 1)$ | $(0.667, 0.942, 1)$ | $(0.5, 0.776, 1)$ | $(0.551, 0.815, 1)$ |
| C33 | $(0.5, 0.776, 1)$ | $(0.667, 0.833, 1)$ | $(0.5, 0.745, 1)$ | $(0.551, 0.783, 1)$ |
| C41 | $(0.5, 0.745, 1)$ | $(0.5, 0.714, 1)$ | $(0.5, 0.684, 1)$ | $(0.5, 0.716, 1)$ |
| C42 | $(0.5, 0.756, 1)$ | $(0.667, 0.833, 1)$ | $(0.5, 0.684, 1)$ | $(0.551, 0.756, 1)$ |
| C43 | $(0.5, 0.756, 1)$ | $(0.333, 0.58, 0.833)$ | $(0.167, 0.362, 0.667)$ | $(0.217, 0.428, 0.717)$ |
| C44 | $(0.167, 0.437, 0.667)$ | $(0.167, 0.425, 0.667)$ | $(0.333, 0.5, 0.667)$ | $(0.22, 0.453, 0.667)$ |
| C51 | $(0.5, 0.76, 1)$ | $(0.333, 0.5, 0.667)$ | $(0.333, 0.58, 0.833)$ | $(0.396, 0.624, 0.846)$ |
| C52 | $(0.333, 0.612, 0.833)$ | $(0.167, 0.362, 0.667)$ | $(0.333, 0.58, 0.833)$ | $(0.283, 0.526, 0.783)$ |

*5.7. Fuzzy Synthetic Operation of Risk Assessment*

In this step, TFNs of all criteria are aggregated based on the fuzzy synthetic evaluation method. Taking "economic risk" as an example, the risk fuzzy synthetic calculation is as follows:

$$h_{c1} = W_{c1} \cdot R_{c1} = (0.2031, 0.3352, 0.0978, 0.3639) \begin{pmatrix} (0.449, 0.677, 0.896) \\ (0.677, 0.882, 1) \\ (0.28, 0.497, 0.73) \\ (0.667, 0.9, 1) \end{pmatrix} = (0.6, 0.828, 0.977)$$

The risk fuzzy calculation results of other indicators are: $h_{c2} = (0.475, 0.699, 0.921)$, $h_{c3} = (0.551, 0.788, 1)$, $h_{c4} = (0.391, 0.609, 0.86)$, $h_{c5} = (0.356, 0.589, 0.824)$.

Then, the overall risk assessment results of the hybrid offshore wind–solar PV power generation project can be calculated as follows:

$$R = W_i \cdot R_c = (0.3145, 0.1903, 0.2501, 0.1519, 0.0932) \begin{pmatrix} (0.6, 0.828, 0.977) \\ (0.475, 0.699, 0.921) \\ (0.551, 0.788, 1) \\ (0.391, 0.609, 0.86) \\ (0.356, 0.589, 0.824) \end{pmatrix} = (0.509, 0.738, 0.94)$$

### 5.8. Defuzzification Process

The overall risk evaluation results are compared with the evaluation items, and the results are between "slightly high" and "high". In order to determine a certain overall risk level, the similarity between the evaluation results and the two items is calculated as follows:

$$
\begin{aligned}
Sd(R, s_4) &= 1 - \frac{\left|R^L - s_4^L\right| + \left|R^M - s_4^M\right| + \left|R^U - s_4^U\right|}{3} \\
&= 1 - \frac{|0.509 - 0.5| + |0.738 - 0.667| + |0.94 - 0.833|}{3} \\
&= 0.938
\end{aligned}
$$

$$
\begin{aligned}
Sd(R, s_5) &= 1 - \frac{\left|R^L - s_5^L\right| + \left|R^M - s_5^M\right| + \left|R^U - s_5^U\right|}{3} \\
&= 1 - \frac{|0.509 - 0.667| + |0.738 - 0.833| + |0.94 - 1|}{3} \\
&= 0.896
\end{aligned}
$$

As can be seen from the above, the overall risk level of the hybrid offshore wind–solar PV power generation project is closer to slightly high. As calculated using the defuzzification process, the risk level of C1 is high, the risk level of C2, C3, C4, is slightly high, and C5 is medium. Figure 7 shows the specific process of the case study.

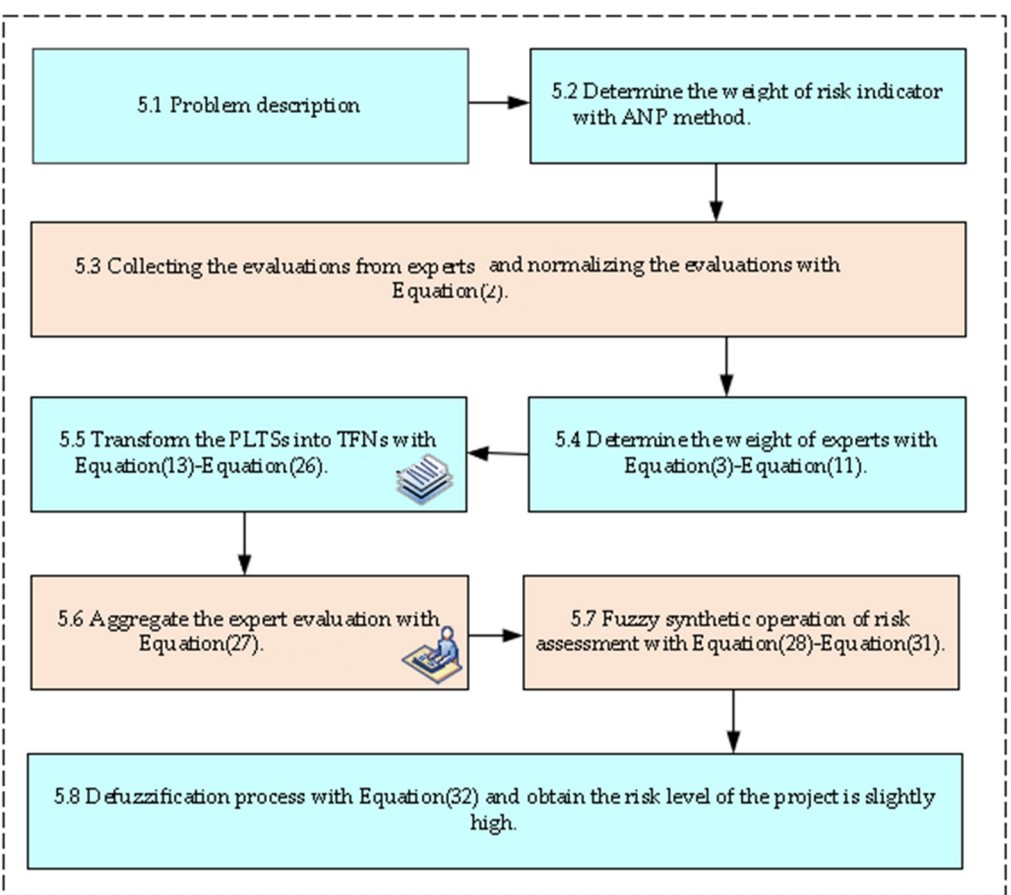

**Figure 7.** The specific process of the case study.

### 5.9. Sensitivity Analysis

In order to test the robustness of the proposed risk assessment framework, the perturbation method was used to conduct a sensitivity analysis on the weight of evaluation criteria; that is, the corresponding changes in the overall risk level of each indicator group and project after the weight of evaluation criteria is slightly disturbed in the decision-making process. $\omega_j$ is the initial weight of the criterion $C_j$, after disturbance it is written as $\omega_j' = \varsigma \omega_j$, where $0 \le \omega_j' \le 1$, and the variation range of parameter $\varsigma$ is $0 \le \varsigma \le 1/\omega_j'$.

According to the normalization of weight, the weight of the other criteria will change accordingly, which is written as $\omega'_k = \phi\omega_k, k \neq j, k = 1, 2, \cdots, m$, and satisfies:

$$\omega'_j + \sum_{k \neq j, k=1}^{m} \omega'_k = 1 \Rightarrow \varsigma\omega_j + \phi \sum_{k \neq j, k=1}^{m} \omega_k = 1 \tag{33}$$

then, $\phi = (1 - \varsigma\omega_j)/(1 - \omega_j)$ is obtained. Firstly, the weight of criteria fluctuated with 20% less and more than the based weight, and the changes in the risk result can be observed intuitively for robustness analysis and sensitive criteria selection. Taking the indicator C12 as an example, we increase the weight by 20%, take $\varsigma = 1.2$, then get the global weight $\omega_{C12} = 0.1265$, the local weight $\omega_{C1i} = (0.1886, 0.3817, 0.091, 0.3386)$, and the weight of the indicator group $\omega_{Ci} = (0.3307, 0.1857, 0.2441, 0.1483, 0.091)$. The triangular fuzzy number is calculated as $R = (0.512, 0.74, 0.941)$. The overall risk level of the project is slightly high, and the risk level of each indicator group has no significant change. When decreasing the weight by 20%, taking $\varsigma = 0.8$, and the calculation process is the same as above. A total of 38 experiments were conducted for each criterion, and the overall risk level results of the project are shown in Table 14. Because the result is represented by a triangular fuzzy number, the change in the result is not obvious. Hence, the sensitivity analysis in this paper is based on similarity measures $Sd(R, s_4)$ with slightly high language term $s_4$. The similarity measure results are shown in Table 15 and Figure 8. It can be seen from Table 15 that the original similarity measure result is $Sd(R, s_4)= 0.938$. When the weight of each indicator changes, the result is relatively stable, indicating that the model has high stability.

**Table 14.** Sensitivity analysis results.

| Indicator | −20% | Initial | +20% |
|---|---|---|---|
| C11 | (0.507, 0.733, 0.933) | (0.509, 0.738, 0.94) | (0.505, 0.73, 0.932) |
| C12 | (0.502, 0.732, 0.951) | (0.509, 0.738, 0.94) | (0.512, 0.74, 0.941) |
| C13 | (0.509, 0.736, 0.937) | (0.509, 0.738, 0.94) | (0.504, 0.731, 0.931) |
| C14 | (0.501, 0.737, 0.93) | (0.509, 0.738, 0.94) | (0.51, 0.736, 0.934) |
| C21 | (0.51, 0.739, 0.941) | (0.509, 0.738, 0.94) | (0.509, 0.737, 0.939) |
| C22 | (0.51, 0.738, 0.941) | (0.509, 0.738, 0.94) | (0.509, 0.737, 0.939) |
| C23 | (0.512, 0.742, 0.945) | (0.509, 0.738, 0.94) | (0.507, 0.734, 0.935) |
| C24 | (0.51, 0.74, 0.943) | (0.509, 0.738, 0.94) | (0.508, 0.735, 0.937) |
| C25 | (0.509, 0.738, 0.94) | (0.509, 0.738, 0.94) | (0.509, 0.738, 0.94) |
| C26 | (0.511, 0.74, 0.942) | (0.509, 0.738, 0.94) | (0.508, 0.736, 0.938) |
| C31 | (0.51, 0.739, 0.941) | (0.509, 0.738, 0.94) | (0.509, 0.737, 0.939) |
| C32 | (0.51, 0.738, 0.942) | (0.509, 0.738, 0.94) | (0.509, 0.736, 0.941) |
| C33 | (0.509, 0.737, 0.939) | (0.509, 0.738, 0.94) | (0.509, 0.736, 0.941) |
| C41 | (0.511, 0.739, 0.94) | (0.509, 0.738, 0.94) | (0.509, 0.737, 0.939) |
| C42 | (0.509, 0.738, 0.941) | (0.509, 0.738, 0.94) | (0.509, 0.737, 0.941) |
| C43 | (0.509, 0.738, 0.94) | (0.509, 0.738, 0.94) | (0.509, 0.733, 0.94) |
| C44 | (0.513, 0.741, 0.943) | (0.509, 0.738, 0.94) | (0.507, 0.735, 0.938) |
| C51 | (0.509, 0.737, 0.939) | (0.509, 0.738, 0.94) | (0.509, 0.738, 0.941) |
| C52 | (0.513, 0.742, 0.944) | (0.509, 0.738, 0.94) | (0.506, 0.734, 0.936) |

**Table 15.** Similarity measure results of the sensitivity analysis.

| Index | C11 | C12 | C13 | C14 | C21 | C22 | C23 | C24 | C25 | C26 |
|---|---|---|---|---|---|---|---|---|---|---|
| −20% | 0.942 | 0.938 | 0.939 | 0.945 | 0.937 | 0.937 | 0.934 | 0.936 | 0.938 | 0.936 |
| +20% | 0.944 | 0.936 | 0.941 | 0.94 | 0.938 | 0.938 | 0.941 | 0.942 | 0.938 | 0.939 |
| Index | C31 | C32 | C33 | C41 | C42 | C43 | C44 | C51 | C52 | |
| −20% | 0.937 | 0.937 | 0.938 | 0.937 | 0.938 | 0.938 | 0.934 | 0.938 | 0.934 | |
| +20% | 0.938 | 0.939 | 0.937 | 0.938 | 0.938 | 0.938 | 0.94 | 0.937 | 0.941 | |

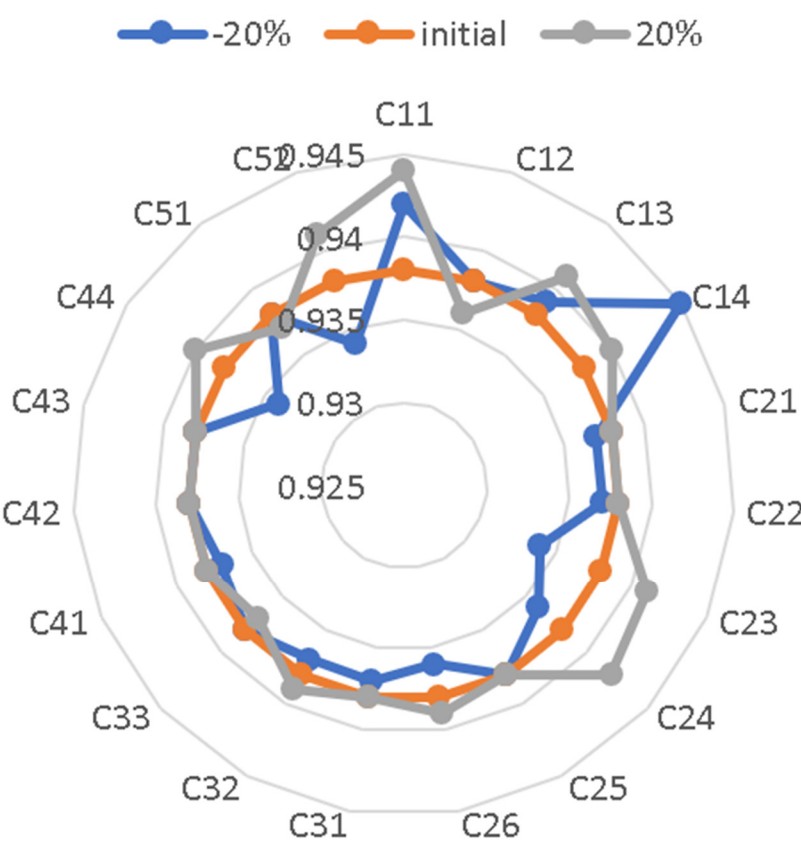

**Figure 8.** The result of the sensitivity analysis.

In Table 15, C11, C14, and C23 are two of the most sensitive indicators. When the weight of these two indicators changes, the results change significantly, but the degree of change is less than 1%. Relevant personnel should pay more attention to operation and maintenance costs and project profitability in the early stage of project construction and take precautions to ensure the smooth implementation of the project. In addition, C25, C42, and C43 are the least sensitive indicators. When the weights of these indicators change, the results remain essentially the same.

*5.10. Comparative Analysis*

The methods proposed by Wu et al. [14], Gao et al. [15], and Jia et al. [39] were applied to verify the rationality and effectiveness of the proposed hybrid offshore wind–solar PV generation project risk assessment framework. Wu et al. [14] established a comprehensive evaluation model. In their work, the hesitant fuzzy linguistic term sets were applied to depict the risk assessment information; ANP was then adopted to calculate the criteria weights; moreover, the evaluations were transformed into the triangular fuzzy sets to facilitate calculation. Gao et al. [15] used probabilistic language terms to evaluate criterion values and proposed a probabilistic linguistic ordered average Choquet integral (PLOAC) operator to aggregate criterion values and deal with the correlation between criteria and the established expert weight determination model based on the PLTS entropy and interaction entropy measures. Jia et al. [39] proposed a risk assessment model integrating the hesitant fuzzy linguistic term sets, triangular fuzzy sets, and eigenvalue method.

In order to make the results comparable, the input risk assessment information is shown in Table 9. In Wu et al. [14], the weight of expert committees is $\omega_i = (0.25, 0.5, 0.25)$ and the weights of indicators are shown in Table 8. In Gao et al. [15], the weight of experts is $\omega_i = (0.3237, 0.339, 0.3372)$ and fuzzy measure was used to calculate the weights of indicators. In Jia et al. [39], the importance degree of each expert is the same, and the method does not consider the correlation between indicators when calculating the weight

of indicators. By using the above methods, the risk level of the project in Wu et al. [14] and Gao et al. [15] is slightly high. The risk level of the project in Jia et al. [39] is medium. Hence, the method proposed in this paper is reasonable. However, there are some differences in the similarity measure results of projects obtained by different methods. Table 16 shows the similarity measure results calculated by the three methods.

**Table 16.** Similarity measure results.

| Method | This Paper | Wu et al. | Gao et al. |
|:---:|:---:|:---:|:---:|
| $Sd(R, s_4)$ | 0.938 | 0.927 | 0.93 |

As can be seen from the table, the results calculated by the method proposed in this paper are slightly different from Wu et al. [14] and Gao et al. [15]. As for Gao et al. [15], the reason why such deviation is produced may be that the subjective and objective characteristics of experts are not considered comprehensively in the determination of expert weight. As for Wu et al. [14], there may be two reasons for this deviation: on one hand, hesitant fuzzy linguistic term sets cannot characterize the probability information of each linguistic term; on the other hand, subjectively assigned expert weight may result in different results. In Jia et al. [39], the overall risk assessment result of the project is medium, and the risk level of C3 and C4 is medium, which has some difference with the method proposed. There may be two reasons for this difference: on one hand, this method does not consider the correlation between indicators, which may lead to inappropriate results; on the other hand, it is assumed that the importance of each expert is equal and each expert has the same weight.

According to the above analysis, the superiorities of this paper are concluded as follows: (i) The probabilistic linguistic fuzzy sets are applied to describe the uncertainties inherent to the risk assessment of the hybrid offshore wind–solar PV generation project. Compared to the hesitant fuzzy linguistic term sets, the probabilistic linguistic evaluation offers detailed probability information for their hesitant evaluation elements, which retains more of the original evaluation information. (ii) The determination of risk indicator weights takes into account the correlation between indicators, which makes the evaluation result more reasonable. (iii) The expert weight determination model adopts the method of combining subjective and objective weights to make the expert weight determination more consistent with reality.

## 6. Discussion

In Section 5, we conducted a case study and calculated that the overall risk assessment result is $R = (0.509, 0.738, 0.94)$ and the risk level of the project is slightly high with the similarity measure result is $Sd(R, s_4) = 0.938$. Through sensitivity analysis and comparative analysis, we can see that the proposed risk assessment framework is robust, scientific, and reasonable. For hybrid offshore wind–solar PV power generation projects, investors in related energy projects should carefully consider whether to invest or not. In addition, it is necessary for the project leader in B city to take effective risk management measures to ensure smooth implementation of the project and reasonable operation. Therefore, this article in view of the general situation of risk management, combining with the characteristics of the hybrid offshore wind–solar PV power generation projects, put forward the corresponding countermeasure and the suggestion, which can provide reference and management implications for policy making and related management personnel.

### 6.1. Economic Risk Countermeasures

High initial investment: Establish a cost control committee and strictly control cost outflow. The combination of economic common sense and technical knowledge can be used to locate and eliminate unnecessary project costs. In addition, BIM software can be used as an auxiliary tool.

High operation and maintenance costs: Use electroplating technology or materials of stainless steel to improve the corrosion resistance of turbines and photovoltaic panel equipment. In addition, professional training of maintenance personnel and real-time monitoring of marine weather conditions to minimize operation and maintenance costs.

Financing risk: Project loans should be implemented before the construction of the project, as much as possible, to use financing channels, disperse funds, and disperse risks.

Profitability risk: Improve project financial management control and arrange financial affairs regularly. In addition, focusing on the profitability and solvency of enterprises and improving the ability of projects to withstand economic risks to ensure the maximization of investment returns.

### 6.2. Technical Risk Countermeasures

Site selection risk: The hybrid offshore wind–solar PV power generation project is a multi-attribute decision-making problem with multiple factors such as resources, economy, and environment. In the early stage of the project, relevant personnel should make full use of the combination of GIS technology and the MCDM method to improve the reliability of project selection and planning and reduce risks as much as possible.

Improper design of hybrid array: Relevant experts were invited to conduct several simulation experiments to select the array design scheme with full utilization of resources and the highest power generation efficiency.

Cable connection risk: Cable connection and integration are a fundamental and important task. People with experience should be invited to participate in the process of designing cable connections. According to ocean tidal effects, cable connections should be made of stronger and more durable materials than onshore energy projects.

System failure risk: In the early stage of the project, several field investigations were carried out to avoid routes such as ship movement and bird migration as much as possible. Digital monitoring technology could be used to predict faults through state monitoring and background data analysis, to minimize the probability of system failures.

Onshore supporting condition risk: Terrain and traffic conditions should be considered during the process of site selection. The project will be hampered by unfavorable traffic conditions. Work roads should be built for large equipment when necessary. Meanwhile, the related personnel of the project should actively communicate with the power grid company to ensure the support for the land power grid.

Visual effect risk: In the process of site selection, the marine wetland ecological area should be avoided as much as possible to avoid affecting the natural beauty. Meanwhile, the reflection degree of photovoltaic panels should be tested in advance to avoid the impact on surrounding residents as much as possible.

### 6.3. Environmental Risk Countermeasures

Wind resource risk: In the process of site selection, wind resource data in the past 60 years should be collected and studied as much as possible. Relevant personnel should conduct field investigations and monitor changes in wind resources to ensure sufficient wind resources at the site of the project and stable power output of the system.

Solar resource risk: In the process of site selection, the solar radiation data of the region should be fully studied, and the annual change and long-term trend of solar resources can be calculated through climate prediction. In addition, in order to ensure the accuracy of data, on-site observation is essential.

Marine ecological damage: Firstly, the location of the hybrid offshore wind–solar PV power generation project should be as far away as possible from marine life and bird habitats, breeding sites, etc. Secondly, during the period of construction, an early warning system for the marine environmental protection should be established to ensure that problems can be resolved in the bud. After the completion of the project, related personnel must apply to the environmental protection department for environmental quality inspection and acceptance.

*6.4. Market Risk Countermeasures*

Market competition risks: It is necessary to focus on innovative technologies to improve the power generation stability of hybrid offshore wind–solar PV power generation projects, to improve market competitiveness and expand market share.

Unclear feed-in tariff policy: We should investigate the power demand and electricity price policy of the project location and surrounding area thoroughly and pay more attention to the documents of governments at all levels and power grid companies related to the electricity price policy of offshore energy projects.

Economic crisis risk: Invite relevant economic experts and set up an expert group. The expert group will study the market based on the hybrid offshore wind–solar PV power generation project and predict its short-term and long-term changes, to take preventive measures in advance.

Human resource shortage risk: Enterprises can establish a human resource management emergency mechanism to ensure the robustness of human resource management in the event of a human resource crisis or emergencies. In addition, enterprises can expand the scope of recruitment of employees, such as college graduates. Enterprises should seize the opportunity, carry out staff training, and strengthen staff theoretical learning and practical operation.

*6.5. Management Risk Countermeasures*

Public opposition risk: In the early stage of the project, the person in charge should actively communicate with the government and surrounding residents and quickly solve and implement the problems and suggestions raised by the residents. Good preparation for the project should be undertaken in the early stage.

Inexperienced staff: Recruit employees with rich experience in offshore energy projects and train them on the combination of theoretical and practical knowledge of hybrid offshore wind–solar PV power projects. Finally, companies should strengthen cooperation with academia to train excellent engineers and project managers.

## 7. Conclusions

In the past 20 years, due to the increasing consumption of fossil fuels, the storage of energy has greatly reduced, and there are some signs of energy shortages. In addition, due to the extensive use of fossil fuels, the discharge of pollutants into the environment has increased. Therefore, reasonable development and utilization of clean energy is present and the future trend of development. In recent years, many new energy power generation projects began to be implemented. Hybrid offshore wind–solar PV power generation projects have attracted much attention for their advantages of saving land resources, high energy efficiency, high power generation efficiency, and stable power output. However, the hybrid offshore wind–solar PV power generation project is still in its initial stage and all aspects are not mature enough; it will inevitably face a series of risks in future investments and construction. Therefore, it is particularly important to establish a related indicator system and a comprehensive risk assessment framework for hybrid offshore wind–solar PV power generation projects.

In this paper, we presented a MCGDM framework for hybrid offshore wind–solar PV power generation projects' risk assessment. Firstly, 19 risk factors were identified and classified into five groups. The ANP method was used to determine the weight of indicators by considering the mutual influence relationship among indicators. Probabilistic linguistic term sets (PLTSs) were then introduced to evaluate the criteria values to depict the uncertainty and fuzziness. Furthermore, the expert weight determination model was built by combining subjective and objective weights. The subjective weighting method was based on the position and project experience of the expert committee and so on, while the objective weighting method was based on the entropy and interaction-entropy measures of PLTSs. In addition, the expert evaluation information was aggregated by transforming PLTSs into triangular fuzzy numbers based on the generalized weighted ordered weighted

averaging (GWOWA) operator. Finally, we presented a case study on the risk assessment of a hybrid offshore wind–solar PV power generation project in B city, and we calculated that the overall risk assessment result is $R = (0.509, 0.738, 0.94)$ and the risk level of the project is slightly high with the similarity measure result $Sd(R, s_4) = 0.938$. The results we calculated can provide certain reference for investors and project managers.

The advantages of the MAGDM framework proposed in this paper are mainly reflected in the following aspects: (1) We established an indicator system for risk assessment of hybrid offshore wind–solar PV power generation projects through literature review, case study, and expert consultation. In total, 19 risk factors of hybrid offshore wind–solar PV power generation projects were determined and they were divided into five groups. (2) PLTS was introduced to describe expert evaluation information. Compared with other forms of fuzzy sets, such as hesitate fuzzy language term set, PLTS can better retain the original evaluation information and make the decision results more credible and reliable. (3) Based on expert information, the entropy measure, and interaction-entropy measure of PLTSs, the expert weight determination model combining subjective and objective information was established to make the decision more in line with reality. (4) The expert evaluation can be transformed into triangular fuzzy number aggregation based on the GWOWA operator, which can minimize the loss and distortion of risk assessment information.

However, there are still some limitations and deficiencies in this paper. Firstly, due to the limited available information, there will inevitably be some omissions in the collection of risk factors. In the future, we can continue to collect information and conduct in-depth research to improve the indicator system. In addition, the risk assessment model and the method of dealing with uncertainty and linguistic variables should be further improved in future research.

**Author Contributions:** Q.M.: Supervision, writing—review and editing, validation. M.G.: Conceptualization, software, data curation, methodology, original draft preparation, visualization. J.L.: Reviewing, English editing, literature survey, and review. J.C.: Reviewing, English editing. P.X.: Supervision, investigation. M.L.: Reviewing. All authors have read and agreed to the published version of the manuscript.

**Funding:** This research was funded by the S&T Program of Hebei (215576116D), Key Research Base Project of Humanities and Social Sciences in Higher Education Institutions of Hebei Province (JJ2109), and Science and technology research and development plan of Qinhuangdao City (202005A068).

**Institutional Review Board Statement:** Not applicable.

**Informed Consent Statement:** Not applicable.

**Data Availability Statement:** Data will be available upon request.

**Acknowledgments:** The authors would like to thank Yanshan University for their support in developing this work.

**Conflicts of Interest:** There is no conflict of interest.

## Nomenclature

| | |
|---|---|
| MAGDM | multi-attribute group decision-making |
| PLTSs | probabilistic linguistic term sets |
| DMs | decision makers |
| GWOWA | generalized weighted ordered weighted averaging |
| PV | photovoltaic |
| MCDM | multiple-criteria decision-making |
| FSE | fuzzy synthetic evaluation method |
| ANP | analytic network process |
| IE | interaction-entropy measure |
| TFN | triangle fuzzy number |
| FPV | floating photovoltaic |

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
