# Peer review of "A Risk Assessment Framework of Hybrid Offshore Wind–Solar PV Power Plants under a Probabilistic Linguistic Environment"

_sustainability, doi:10.3390/su14074197_

Round 1
Reviewer 1 Report
The manuscript describes the risk assessment of offshore hybrid power plant installations. The manuscript clearly describes the risk assessment methodology used.
- The authors presented a theoretical introduction concerning both the technical background related to the hybrid power plant technique and the theory of risk analysis. The verbal description is also graphically aided, eg. Fig 1.
- The manuscript describes the experiment in detail and gives the relevant scientific theory. The experiment is also graphically described. eg. Fig 2.
- As part of Chapter 5 of Case study, an experiment was recalled that was described in detail. Readers can trace all the data and make analysis for their own needs.
- The conclusions are sufficient for me. The authors commented on the results in Chapter 6.
- Captions under the figures should be corrected
- On pages 19 and 22 the pictures have the same number 4 . If possible, improve the legibility of the drawing on page 22 .
- The graphic side of the manuscript should be improved
- As for me, Marine environment risk (C33): in terms of corrosion, it is negligible and should not be analyzed. It is obvious that all marine installations are made in accordance with the technology, with appropriate anti-corrosion protection.
Author Response
Dear Reviewer#1:
Thank you for your comprehensive and insightful comments on our manuscript entitled "Rask Assessment of Hybrid Offshore Wind-Solar PV Power Plants under Probabilistic Linguistic Environment". We have studied the comments carefully which are extremely helpful for revising and improving our paper and made corrections that we hope meet with approval. According to the invaluable comments, suggestions and questions from reviewer, we have completed the revision of our manuscript. The revised portions are highlighted in red in the paper and the main revisions in light of your comments are as follows. Please see the attachment.

Reviewer 2 Report
I congratulate the authors for their paper entitled "Risk Assessment of Hybrid Offshore Wind-Solar PV Power 2 Plants under Probabilistic Linguistic Environment." in which they identify 19 risk factors and they are used to determine, by fuzzy synthetic evaluation method, 20 and similarity measure and give the corresponding risk response strategy.
Even though I find the overall article as being fluent and well developed, more information on the following sections should be added:
- The Title - should be a title, not a sentence
- The Abstract - it should not contain abbreviations, as they will be mentioned in the corpus of the paper
- The Literature review should be extended with more references that are both international and local, presenting also the used methods, but also the current state of art
- Explain and broaden the information related to the choice for the variables. The data needs more explanations. (you are saying there is a shortage of literature but at the methodology you say "literatures published by domestic and foreign scholars are searched by keywords such as offshore wind power, offshore photovoltaic and offshore new energy projects" ... is there a literature in the field or not?); "the expert committee determined the risk criteria system of hybrid offshore wind-solar PV power generation project through brainstorming" - you took their ideas for granted?)
- Add a new section named Discussion in which you explain the findings, you draw recommendations and then you have the Conclusions part
- In the paper, you are using a lot "Firstly .... ". The following mentioned idea should be starting with "Secondly". Avoid the use of this phrase if you do not count
Some other recommendations:
- The Figures and Tables should be written according to MDPI's template (and their explanations in text) - you have 2 times Figure 4
- Why do you have a nomenclature, when you explained the abbreviations in the text?
Author Response
Dear Reviewer#2:
Thank you for your comprehensive and insightful comments on our manuscript entitled "Rask Assessment of Hybrid Offshore Wind-Solar PV Power Plants under Probabilistic Linguistic Environment". We have studied the comments carefully which are extremely helpful for revising and improving our paper and made corrections that we hope meet with approval. According to the invaluable comments, suggestions and questions from reviewer, we have completed the revision of our manuscript. The revised portions are highlighted in red in the paper and the main revisions in light of your comments are as follows. Please see the attachment.

Reviewer 3 Report
The article is devoted to building a multi-criteria group decision-making structure for hybrid offshore wind-solar projects to produce photovoltaic energy. The study's relevance is dictated by the fact that hybrid offshore wind-solar photovoltaic projects show the advantages of land saving, high energy efficiency, high power generation efficiency, and stable power output in recent years. The authors identify 19 risk factors, which are classified into five groups. Then, probabilistic linguistic term sets are introduced to evaluate the values of the criteria for displaying uncertainty and fuzziness. A model for determining expert weight is built by combining subjective and objective weights based on expert information, entropy and entropy-interacting indicators, and probabilistic linguistic sets of terms. In addition, expert judgment information is aggregated by converting probabilistic linguistic term sets into fuzzy triangular numbers based on the generalized weighted ordered weighted averaging operator. The level of risk is determined by the fuzzy synthetic evaluation method and the measure of similarity and gives the appropriate risk response strategy. The system of risk indicators and corresponding countermeasures proposed by the authors provide a scientific and theoretical basis for making investment decisions and avoiding the risks of hybrid offshore wind-solar photovoltaic projects.
Despite the satisfactory quality of the article, some shortcomings need to be corrected.
- The abstract should contain numerical results obtained by the authors.
- The aim of the article should be defined.
- It is recommended to consider including the risk of human resource shortages relevant in the context of the global COVID-19 pandemic.
- Figures captures should begin with capital letters.
- The conclusion section should contain numerical results obtained by the authors.
- The scientific and practical novelty of the paper should be highlighted.
- It is recommended to include formal transformation approaches in multiple-criteria decision-making literature review, e.g., doi: 10.3390/en14248235
In summarizing my comments, I recommend that the manuscript be accepted after minor revision.
Author Response
Dear Reviewer#3:
Thank you for your comprehensive and insightful comments on our manuscript entitled "Rask Assessment of Hybrid Offshore Wind-Solar PV Power Plants under Probabilistic Linguistic Environment". We have studied the comments carefully which are extremely helpful for revising and improving our paper and made corrections that we hope meet with approval. According to the invaluable comments, suggestions and questions from reviewer, we have completed the revision of our manuscript. The revised portions are highlighted in red in the paper and the main revisions in light of your comments are as follows. Please see the attachment.

Round 2
Reviewer 2 Report
I congratulate the authors for revising their research, significantly improving their paper, and addressing all my comments. I have no additional comments or recommendations.